# Tunable cell differentiation via reprogrammed mating-type switching

Yu Chyuan Heng [1,2], Shohei Kitano[1,2,3,4], Adelia Vicanatalita Susanto[1,2,3,4], Jee Loon Foo [1,2,3,4] ✉ & Matthew Wook Chang [1,2,3,4] ✉

This study introduces a synthetic biology approach that reprograms the yeast mating-type switching mechanism for tunable cell differentiation, facilitating synthetic microbial consortia formation and cooperativity. The underlying mechanism was engineered into a genetic logic gate capable of inducing asymmetric sexual differentiation within a haploid yeast population, resulting in a consortium characterized by mating-type heterogeneity and tunable population composition. The utility of this approach in microbial consortia cooperativity was demonstrated through the sequential conversion of xylan into xylose, employing haploids of opposite mating types each expressing a different enzyme of the xylanolytic pathway. This strategy provides a versatile framework for producing and fine-tuning functionally heterogeneous yet isogenic yeast consortia, furthering the advancement of microbial consortia cooperativity and offering additional avenues for biotechnological applications.

Microbial biotechnology has predominantly been centered on the engineering of single-cellular organisms, a focus that inherently constrains the complexity of achievable functionalities[1]. Contrasting with this, evolutionary processes have favored the emergence of microbial consortia as a means to undertake complex biological functions. These consortia are characterized by cooperative interactions among specialized individuals within a collective, highlighting the evolutionary advantage of such complexity[2–4]. This pattern of synergistic and mutualistic interactions is not isolated but widespread across nature, indicating their critical role in enhancing the physiological and ecological complexity of microbial communities[5]. Leveraging nature's designs, researchers have begun harnessing microbial consortia for cooperative applications, such as distributing biosynthetic pathways across different cell types to enhance productivity while minimizing metabolic burden and separating incompatible processes[1,6,7]. However, the reliance on manual mixing of different cellular platforms introduces challenges in the formation of microbial consortia, limiting the broader application of this strategy, especially in scenarios where direct human intervention is not feasible. Bioengineering strategies

have sought to overcome these limitations by engineering genetic circuits that enable the formation of microbial consortia via asymmetric cell division and differentiation[8,9]. Despite this progress, inducible cell differentiation in eukaryotic models, such as *Saccharomyces cerevisiae*, remains largely unexplored. Nonetheless, haploid *S. cerevisiae* naturally divides and differentiates asymmetrically, presenting an opportunity to harness these underlying mechanisms for tunable cell differentiation to facilitate synthetic microbial consortium formation and cooperativity.

The processes of asymmetric cell division and differentiation in haploid *S. cerevisiae* are highly coordinated events. While the asymmetric budding of a smaller haploid daughter cell from a larger haploid mother cell is common across yeast strains, the asymmetric differentiation process enabling a haploid mother cell to switch mating type is unique to homothallic yeast strains with a dominant *HO* gene (Fig. 1a). The *HO* gene encodes a site-specific endonuclease essential for the mating-type switching process, facilitating sexual differentiation by initiating a directional DNA repair mechanism that interconverts the mating-type allele at the *MAT* locus[10–12] (Fig. 1b). The *MAT*

[1]Department of Biochemistry, Yong Loo Lin School of Medicine, National University of Singapore, Singapore, Singapore. [2]NUS Synthetic Biology for Clinical and Technological Innovation (SynCTI), National University of Singapore, Singapore, Singapore. [3]Synthetic Biology Translational Research Programme, Yong Loo Lin School of Medicine, National University of Singapore, Singapore, Singapore. [4]National Centre for Engineering Biology (NCEB), Singapore, Singapore. ✉e-mail: jeeloon.foo@nus.edu.sg; bchcmw@nus.edu.sg

locus houses either the α- or a-mating-type allele but only the transcriptional factors (α1 and α2) encoded by the former are involved in the regulation of mating-type-specific gene expression. A switch in the allele at the *MAT* locus thus changes the mating type of haploids[10,11,13,14]. Naturally, the asymmetry in mating-type switching is crucial in ensuring the concurrent presence of haploids of opposite mating types, facilitating self-fertilization to convert haploids to the advantageous diploid form[10,15]. Specifically, mating-type switching is programmed to occur in the older haploid mother cell only after it has divided, producing a smaller haploid daughter cell that retains the original mating type[10,16]. This process is achieved mechanistically by restricting the translation of the Ash1 transcriptional repressor and its subsequent inhibition of *HO* expression, to the haploid daughter cell through the asymmetric transport of its mRNA to the distal bud tip[17–20]. In contrast, the depletion of Ash1 in the haploid mother cell enables *HO* expression, following sequential activation by the Swi5 and SBF (Swi5/Swi6) transcriptional activators[21,22]. The intricate process of mating-type switching led us to hypothesize its potential to be rewired into a genetic circuit for tunable cell differentiation to facilitate synthetic microbial consortium formation and cooperativity.

Here, we present a synthetic biology approach that repurposes the yeast mating-type switching mechanism into a genetic logic gate capable of asymmetric cell differentiation, generating tunable microbial consortia comprising two isogenic but phenotypically and functionally distinct haploids of opposite mating types. As a proof-of-concept, we demonstrated the utility of this approach in facilitating microbial consortia cooperativity, by distributing the expression of xylanolytic enzymes across two haploids of opposite mating types for the sequential conversion of xylan into xylose. This study provides a versatile tool for producing and fine-tuning functionally heterogeneous, yet isogenic yeast consortia characterized by mating type heterogeneity, contributing significantly to the advancement of microbial consortia cooperativity.

## Results

### Designing mating-type switching circuity
To repurpose mating-type switching in haploid *S. cerevisiae* for tunable cell differentiation to facilitate synthetic microbial consortium formation and cooperativity, we proposed that artificial control could be established through direct or indirect modulation of *HO* expression, while phenotypic differences could be introduced via differential gene expression regulated by mating-type-specific promoters. A prerequisite involves disrupting the native mating mechanism to prevent diploid formation, which would otherwise suppress the expression of mating-type-specific and haploid-specific genes, including *HO*. The resulting sterile haploid strain would undergo asymmetric *HO* expression and mating-type switching in every cell cycle, thus forming a microbial consortium with haploids of opposite mating types maintained at a 1:1 ratio. Although this system provides stability, it faces challenges in generating diverse population compositions. To address this, we selected the yeast strain BY4742 as the base strain for engineering. Notably, this α-mating type haploid strain carries a T-to-A "stuck mutation," which decreases the a-to-α mating type conversion by 70–90%[23,24]. This alteration biases the overall switching in the direction of α-to-a mating-type, thus, offering an opportunity to fine-tune population composition by adjusting the degree of mating-type switching with variable inducer concentrations.

### Establishing a sterile base strain with a fluorescent reporter system
To generate a sterile haploid yeast strain, we deleted genes critical for mating signaling, specifically the mating pheromone and pheromone receptor genes. In the α-mating type haploid BY4742, the α-factor mating pheromone is encoded by the *MFα1* and *MFα2* genes, while the a-factor pheromone receptor is encoded by the *STE3* gene[11,25]. Deletion

of the cryptic *MFa1*, *MFa2*, and *STE2* genes encoding the a-mating-type equivalents in the BY4742 strain was deemed unnecessary, as rendering one mating type defective is sufficient to sterilize a population[26–30]. Consequently, we constructed single-, double-, and triple-gene deletion strains of *MFα1*, *MFα2*, and *STE3* genes (MTS001–MTS007) and crossed them with the a-mating-type haploid BY4741 strain. Sterility was assessed based on auxotrophy on lysine and methionine drop-out plates that were selected only for diploids formed from the conjugation of BY4742 derivatives (Lys⁻, Met⁺) and BY4741 (Lys⁺, Met⁻). As shown in Fig. 1c, a single deletion of the *STE3* gene and double deletions of the *MFα1* and *MFα2* genes effectively disrupted mating. However, to avoid potential growth disparities stemming from pheromone-induced $G_1$-arrest[11] that could affect the stability of a microbial consortium, we selected the triple-gene deletion strain MTS007, which has the complete set of *MFα1*, *MFα2*, and *STE3* genes disrupted, as our sterile base strain.

Subsequently, we developed a reporter system based on in vivo conditional mating-type-specific expression of fluorescent proteins, enabling precise determination of mating types at the single-cell level and quantification at the population level using flow cytometry. Specifically, *GFP* and *mCHERRY* genes were integrated into the *mfα1Δ* and *MFa1* loci, respectively, of the sterile α-mating type haploid strain, MTS007, to create strain MTS008α. An a-mating type equivalent, MTS008a, was also constructed. As the mating-type-specific promoters remained intact, the expression of *GFP* and *mCHERRY* genes was theoretically restricted to α- and a-mating type haploids, respectively. As illustrated in Fig. 1d and S1, GFP was exclusively expressed in the α-mating type haploids (MTS008α), and mCherry in the a-mating type haploids (MTS008a). Notably, red fluorescence higher than the BY4742 control but lower than the MTS008a strain was observed in the MTS008α strain, which was later found to be caused by a spillover signal from GFP detected by the red fluorescence channel (FL4). Both MTS008α and MTS008a strains formed distinct populations with minimal overlap on a fluorescence scatter plot. However, direct quantification of population composition by evaluating the proportion of cells in Q3 and Q4 was challenging due to the presence of a small subpopulation of α-mating type haploids (~2.63% based on 144 samples) that overlapped with the a-mating type haploids in Q4. To account for this, we estimated the proportion of α-mating type haploids in a population by dividing the percentage of cells in Q3 by 97.37% and determined the proportion of a-mating type haploids by subtracting the calculated percentage of α-mating type haploids from 100%.

### A YES logic gate based on direct modulation of *HO* expression
After establishing a sterile base strain equipped with fluorescent reporters, we next aimed to artificially control the yeast mating-type switching mechanism. Notably, the formation of a consortium with mating-type heterogeneity is typically unfeasible through direct modulation of *HO* expression in the wild-type haploid *S. cerevisiae* strains. This is because replacing the *HO* promoter would disrupt the critical spatiotemporal pattern of *HO* expression necessary for establishing sexual asymmetry between the haploid mother and daughter cells. However, this might be viable with the BY4742 strain, which exhibits a deficiency in a-to-α mating type conversion. While the non-selective nature of the replacement promoter will initially cause both the haploid mother and daughter cells to switch from α-to-a mating type, the subsequent conversion from a-to-α mating type will only occur in a portion of the population. Consequently, this results in a microbial consortium comprising haploids of opposite mating types, where the composition can be adjusted by modulating the net α-to-a mating type conversion through the regulation of *HO* expression from the replacement promoter (Fig. 2a).

To test this hypothesis, we replaced the *HO* promoter and the recessive *ho* gene in the MTS008α strain with the inducible *GAL1*

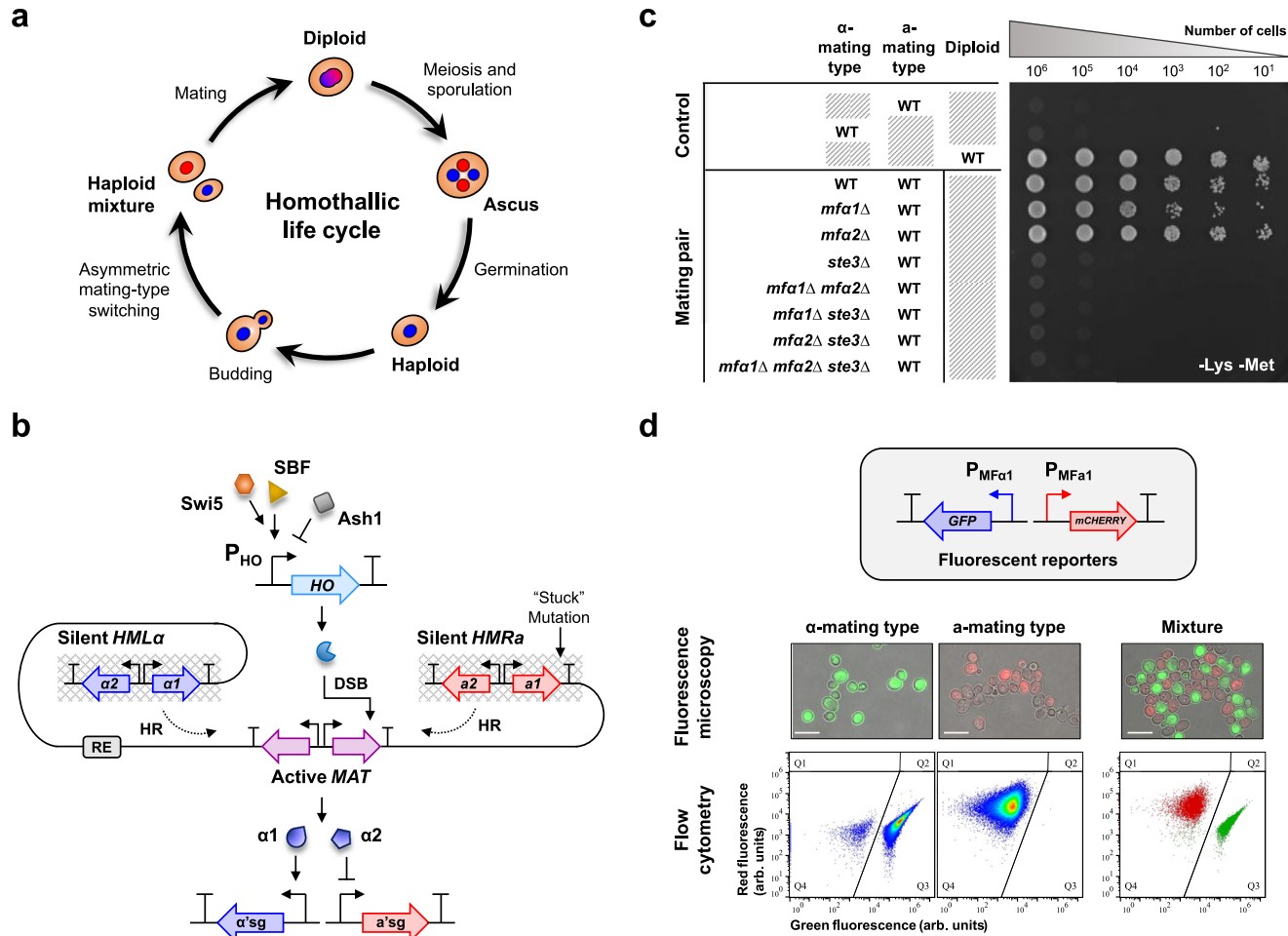

**Fig. 1 | Mating-type switching and mating-type-specific gene expression. a** A typical homothallic life cycle of *S. cerevisiae*. **b** The mating type of a haploid is governed by the active *MAT* locus, which houses either the α- or a-mating-type alleles. The α-mating-type allele encodes the transcriptional factors α1 and α2, which are respectively involved in the activation of α-mating-type-specific genes (α'sg) and the repression of a-mating-type-specific genes (a'sg) in α-mating type haploids. In a-mating type haploids, the expression of a'sg occurs in the absence of α2 and does not involve gene products (a1 and a2) of the a-mating-type allele. Mating-type switching is initiated upon the introduction of a double-stranded break (DSB) by the HO endonuclease, which triggers a directional DNA repair mechanism at the *MAT* locus mediated by the recombinational enhancer (RE). By using the opposite mating-type allele at the silent *HMLα* or *HMRa* locus as the donor template for homologous recombination (HR), this process interconverts the mating-type allele at the *MAT* locus. *HO* expression is activated by the sequential actions of the Swi5 and SBF (Swi5/Swi6) transcriptional activators and repressed by the Ash1 transcriptional repressor. The haploid yeast strain BY4742 used in this study possesses a T-to-A "stuck mutation" at the silent *HMRa* locus. **c** Generation of sterile α-mating type haploid BY4742 strains (Lys⁻, Met⁺) by deleting the *MFα1-2* and *STE3* genes. Sterility was examined by mating with the a-mating type haploid BY4741 (Lys⁺, Met⁻) and inferred as a growth defect on the lysine and methionine drop-out plate that selected only diploids (Lys⁺, Met⁺). Diploid BY4743 was included as a control. **d** Development of a fluorescent reporter system by integrating the *GFP* and *mCHERRY* genes into the *mfα1Δ* and *MFα1* loci, respectively. The expression of *GFP* and *mCHERRY* was mutually exclusive, with the former specific to the α-mating type haploids (green; represented by MTS008α strain) and the latter to the a-mating type haploids (red; represented by MTS008a strain). Scale bars on the micrographs represent 10 μm. The micrographs shown are representative of *n* = 3 biological replicates.

promoter and the dominant *HO* gene, by integrating the P$_{GAL1}$-*HO*-T$_{CYC1}$ expression module into the recessive *ho* locus. The resulting strain, MTS009, was incubated for one day in a medium containing either the repressor glucose or the inducer galactose. Subsequent flow cytometry measurements were conducted to examine the population composition. As shown in Fig. 2b and S2a, contrary to a near-homogeneous population predominantly composed of 99.7% α-mating type haploids in the medium with 2% glucose, inducing *HO* expression with 2% galactose resulted in a consortium comprising 12.3% α-mating type haploids and 87.7% a-mating type haploids. Further alterations in galactose concentration from 0.5% to 8.0% did not significantly change the population composition (slope = 0.57; $R^2$ = 0.84), which we conjectured reached equilibrium with the α-to-a mating type ratio maintained at ~0.14.

Although limited to an equilibrium population skewed towards the a-mating type haploids, this YES logic gate nonetheless demonstrates the potential for generating a microbial consortium through direct modulation of *HO* expression. Its inability to generate a diverse range of population compositions suggests that the degree of *HO* expression and α-to-a mating type conversion should be further reduced. Indeed, this could potentially be accomplished by using the *HO* promoter, which restricts mating-type switching to the haploid mother cell only, affecting half of the population. This unique P$_{HO}$-*HO* expression pattern acts as a natural dilution strategy, a feature we aimed to preserve in our subsequent system designs.

**A YES logic gate based on indirect modulation of *HO* expression**

The retention of the P$_{HO}$-*HO* expression module necessitated an alternative target for modulation. Obvious candidates included the upstream transcriptional factors involved in *HO* regulation: the activators Swi5 and SBF, and the repressor Ash1. In the native system, Swi5 and SBF sequentially activate *HO* expression in the haploid

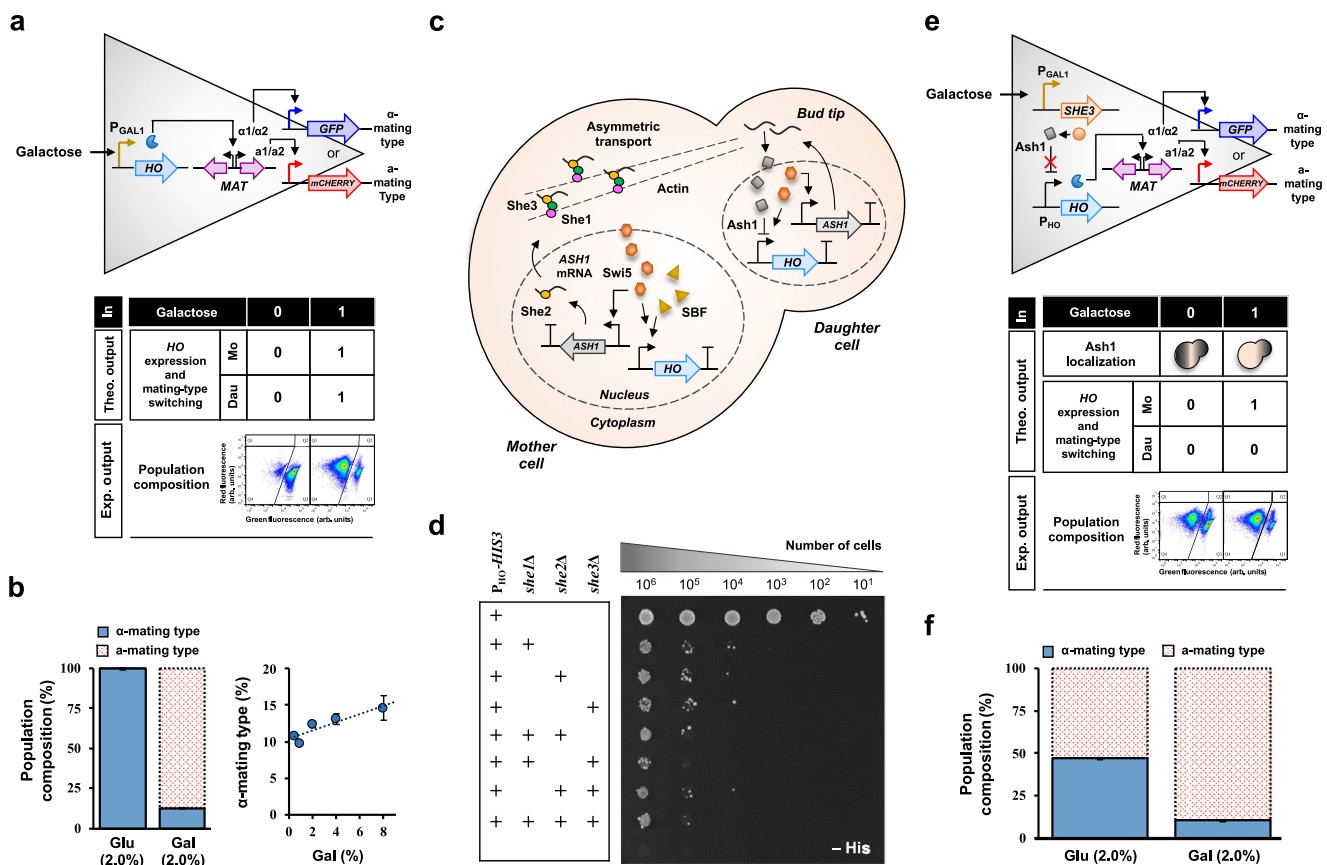

**Fig. 2 | YES logic gates for modulating *HO* expression to form microbial consortia. a** Development and **b** characterization of a YES logic gate using a *GAL1* promoter for direct modulation of *HO* expression. In this system, replacing the *HO* promoter eliminated the asymmetry between mother and daughter cells. *HO* expression and mating type switching in both cell types were activated by galactose and repressed by glucose. A microbial consortium formed upon galactose induction in strain MTS009 integrated with P_GAL1-*HO*-T_CYC1 module. *n* = 6 biological replicates. **c** The *ASH1* mRNA asymmetric transport system. The *ASH1* mRNA is bound by She2 co-transcriptionally to form a messenger ribonucleoprotein particle (mRNP) prior to nuclear exit. In the cytoplasm, the mRNP is further bound by She1 and She3 to form a mature complex that moves asymmetrically along the actin filaments to the distal bud tip. This results in asymmetric Ash1 translation, inhibiting *HO* expression in the daughter cell. Ash1 depletion in the mother cell permits *HO* activation by Swi5 and SBF transcriptional factors. **d** A semi-quantitative genetic assay using a P_HO-*HIS3*-T_HO reporter module to examine the impact of *SHE1-3* gene deletion on *ASH1* mRNA asymmetric transport. Disruption to the transport system was inferred from growth defects on histidine drop-out plates. **e** Development and **f** characterization of a YES logic gate based on indirect modulation of *HO* expression, by regulating *SHE3* expression using a *GAL1* promoter. In this design, *SHE3* inhibition in the presence of glucose disrupted the *ASH1* mRNA asymmetric transport system, leading to symmetric *HO* inhibition in both mother and daughter cells. In contrast, *SHE3* activation in the presence of galactose reinstated the asymmetry in *HO* inhibition, generating a microbial consortium in which only the mother cell switches mating type. Experimentally, a microbial consortium formed even in the absence of galactose in strain MTS018 integrated with the P_GAL1-*SHE3*-T_CYC1 module, due to leaky *HO* expression. *n* = 3 biological replicates. Values shown in panels **b** and **f** represent the mean ± standard deviation. Mo mother cell, dau daughter cell, glu glucose, gal galactose. Source data are provided as a Source Data file.

mother cell by initiating chromatin remodeling events that expose the otherwise inaccessible *HO* promoter for binding with the general transcriptional machinery[21,22]. Conversely, asymmetric translation of Ash1, following the active relocalization of its mRNA from the haploid mother cell, results in asymmetric *HO* inhibition in the haploid daughter cell through repressive chromatin formation[17–20,22]. However, given their roles as generic cell-cycle-dependent transcriptional factors[31–33], directly modulating *SWI5*, *SBF*, and *ASH1* gene expression was deemed inappropriate due to the potential detrimental effects of their overexpression or absence. Instead, we hypothesized an indirect approach to control *HO* expression by modulating the activity of the *ASH1* mRNA asymmetric transport system. Disrupting this transport system would lead to symmetric Ash1 localization and *HO* inhibition in both the haploid mother and daughter cells. Conversely, restoring the transport system would reinstate asymmetric Ash1 localization and *HO* inhibition in the haploid daughter cell only, enabling the haploid mother cell to switch mating types and subsequently generating a microbial consortium comprising haploids of opposite mating types.

The *ASH1* mRNA asymmetric transport system involves multiple proteins (Fig. 2c). Among these, the roles of She1-3 proteins were most prominent, as mutations in these proteins were previously shown to result in *ASH1* mRNA and protein distribution defect[18,34]. She2 is an RNA-binding protein that binds *ASH1* mRNA co-transcriptionally in the nucleus to form a messenger ribonucleoprotein particle (mRNP). She1 and She3 are a myosin motor and an adaptor protein, respectively, that bind the mRNP upon its nuclear exit to form a mature cytoplasmic transport complex that moves asymmetrically along the actin filaments to the distal bud tip[35,36]. Given their significance, we screened *SHE1-3* genes as potential modulation candidates for the *ASH1* mRNA asymmetric transport system using a semi-quantitative genetic assay. Single-, double-, and triple-gene deletion mutants were constructed and integrated with a P_HO-*HIS3*-T_HO reporter module (MTS010−MTS017). A suitable modulation candidate was selected based on its deletion disrupting the *ASH1* mRNA asymmetric transport system, leading to *HO* inhibition in both the haploid mother and daughter cells. This disruption was inferred from growth defects on

histidine drop-out plates due to the inhibition of *HIS3* expression from the *HO* promoter.

As shown in Fig. 2d, single-, double-, and triple-gene deletions of *SHE1-3* genes resulted in comparable growth defects on the histidine drop-out plate, suggesting that a single-gene deletion is sufficient to disrupt the *ASH1* mRNA asymmetric transport system. However, the inhibition of *HIS3* expression from the *HO* promoter was not absolute, as some cell growth was still observed on the plate. We hypothesized that this incomplete inhibition was not due to a partial disruption of the *ASH1* mRNA asymmetric transport system; otherwise, we would expect less cell growth with an increasing number of deletions. Instead, these incomplete inhibitions common to all deletion mutants likely resulted from leaky *HIS3* expression from the *HO* promoter. Indeed, when the *HIS3* reporter gene was replaced with the *GFP* gene, similar leaky expression from the *HO* promoter was observed, resulting in green fluorescence approximately two-fold (two-tailed Student's *t*-test, *P* value ≤ 0.0002) higher than that of the BY4742 control and the strain MTS030 with repressed *GFP* expression from the *GAL1* promoter (Fig. S3). The *SHE3* gene was selected as the modulation target for the *ASH1* mRNA asymmetric transport system because its mutation (0.7%) was previously reported to yield the least amount of leaky mating-type switching in the mother cells as compared to the *she1* (1.5%) and *she2* (5.5%) mutants[34].

We next artificially modulated the activity of the *ASH1* mRNA asymmetric transport system by regulating *SHE3* expression using the inducible *GAL1* promoter (Fig. 2e). For this, we replaced the recessive *ho* and *SHE3* genes in the MTS008α strain with the dominant *HO* gene and the P$_{GAL1}$-*SHE3*-T$_{CYC1}$ expression module, respectively. The resulting strain, MTS018, was assessed for population composition using a flow cytometer after one day of incubation in a medium containing either the repressor glucose (2%) or the inducer galactose (2%). Notably, a subpopulation was observed in Q3 of the fluorescence scatter plot, representing haploids exhibiting high green and red fluorescence (Fig. 2e). This subpopulation was also present in strain MTS009 after galactose induction, although less prominently (Fig. 2a). Considering the differential kinetics of the mCherry and GFP fluorescent proteins, we inferred that this subpopulation comprised haploids that recently switched from a-to-α mating type. Due to a longer half-life[37], mCherry remains in the α-mating type haploids, leading to its concurrent presence with GFP, which has a shorter maturation time. This scenario is unlikely when the haploids switch from α-to-a mating type, as the degradation of GFP is faster than the maturation of mCherry.

A microbial consortium consisting of 46.8% α-mating type haploid and 53.2% a-mating type haploids was formed in the medium containing the repressor glucose (Fig. 2f and S2b), attributable to leaky *HO* expression from the *HO* promoter as observed earlier (Fig. 2d and S3). The proportion of α-mating type haploids further dropped from 46.8% to 10.5% upon galactose induction. Although this signified the successful development of a YES logic gate based upon the conditional activation of asymmetric *HO* expression and sexual differentiation for microbial consortium formation, the occurrence of leaky mating-type switching during non-inducing conditions called for a strategy to tighten *HO* expression.

## An AND logic gate for tunable synthetic microbial consortium composition control

To enhance the regulation of *HO* expression, we substituted the leaky *HO* promoter with a hybrid promoter developed in a recent study[38]. This modification involved replacing the 145 bp sequence upstream of the *HO* gene with the 197 bp core promoter sequence of the *TX* promoter, which contains a tandem pair of *tetO* sites located downstream of the TATA box. Basal *HO* inhibition was established through the constitutive expression of TetR, which binds to the *tetO* sites and prevents RNA polymerase from binding. This inhibition

was implemented by integrating the P$_{TEF1}$-*TetR*-T$_{CYC1}$ expression module into the *mfα2Δ* locus of the MTS018 strain. Lifting the basal *HO* inhibition necessitated the addition of tetracycline to dissociate TetR from the *tetO* sites. As such, substitution to this hybrid *HO* promoter introduced an extra layer of regulation, transforming the initial YES logic gate into an AND logic gate co-regulated by both tetracycline and galactose inducers (Fig. 3a). The upstream regulatory sequences URS1 (−1900 to −1000) and URS2 (−900 to −200) of the *HO* promoter[39], critical for maintaining asymmetry in *HO* expression, were unaltered in this construct. Consequently, *HO* expression, following de-repression by tetracycline, was similarly confined to the haploid mother cell upon restoration of the *ASH1* mRNA asymmetric transport system through galactose-induced *SHE3* expression.

The resulting strain, MTS019, formed a near-homogeneous population predominated by 98.8% α-mating type haploids in a medium containing 2% glucose repressor (Fig. 3b and S2c). This indicated a significant reduction in leaky *HO* expression attributed to the basal *HO* inhibition imposed by the TetR-*tetO* pair. This basal inhibition also affected mating-type switching in a medium containing 2% galactose inducer, resulting in a population comprising 97.0% α-mating type haploids. However, when 25 µg mL⁻¹ tetracycline was added to the galactose-containing medium, ~57.7% of the population switched mating types. Notably, lifting the basal *HO* inhibition did not lead to an increase in leaky *HO* expression in the medium containing 25 µg mL⁻¹ tetracycline and 2% glucose, which formed a population with 93.7% α-mating type haploids. This effectively demonstrated the establishment of a tightly regulated AND logic gate, activated only in the presence of both tetracycline and galactose inducers, for forming a microbial consortium.

We then explored the range of population compositions achievable by the AND logic gate using different combinations of inducer concentrations. Initially, we quantified the logic gate's response to increasing tetracycline concentrations in a medium containing 2% galactose. A linear decrease in the proportion of α-mating type haploids was observed at concentrations ranging from 6.25 to 25.0 µg mL⁻¹ tetracycline (slope = −2.97; $R^2$ = 0.99), as shown in Fig. 3c. Adjusting the galactose concentration from 0.5% to 8.0% yielded similar linear dose responses, with a more pronounced change occurring in the medium with 12.5 µg mL⁻¹ (slope = −4.34; $R^2$ = 0.99) than with 25.0 µg mL⁻¹ (slope = −1.33; $R^2$ = 0.92) tetracycline (Fig. 3d). These varied responses to different inducer concentrations demonstrated the tunability of the AND logic gate, which produced a diversity of population compositions with α-to-a mating type ratios ranging from 0.5 to 81.4 (Fig. 3e). This corresponds to microbial consortia containing a minimum of 32.9% α-mating type haploids and a maximum of 67.1% a-mating type haploids, or up to 98.8% α-mating type haploids and a minimum of 1.2% a-mating type haploids.

The stability of the microbial consortia formed was assessed by monitoring changes in population composition after the removal of inducers. As shown in Fig. 3f, all five consortia, established after one day of induction in a medium with varying inducer concentrations (12.5 or 25.0 µg mL⁻¹ tetracycline with 0.5%, 1.0%, or 8.0% galactose, or 2% glucose), consistently maintained their distinct population compositions for up to four days of repassaging (diluted 20-fold) in a fresh medium containing 2% glucose repressor. This indicated the stability of the consortia in the absence of inducers. Interestingly, this stability in population composition persisted even when the consortia were cultivated for an additional two days in the same inducing medium (Fig. 3g). In addition to the depletion of one or both inducers, particularly galactose, which serves as a carbon source for the cells in the absence of glucose[40,41], we postulated that stability in this scenario could result from cell saturation after one day of cultivation, which could limit further cell divisions necessary for mating-type switching. To investigate further, we

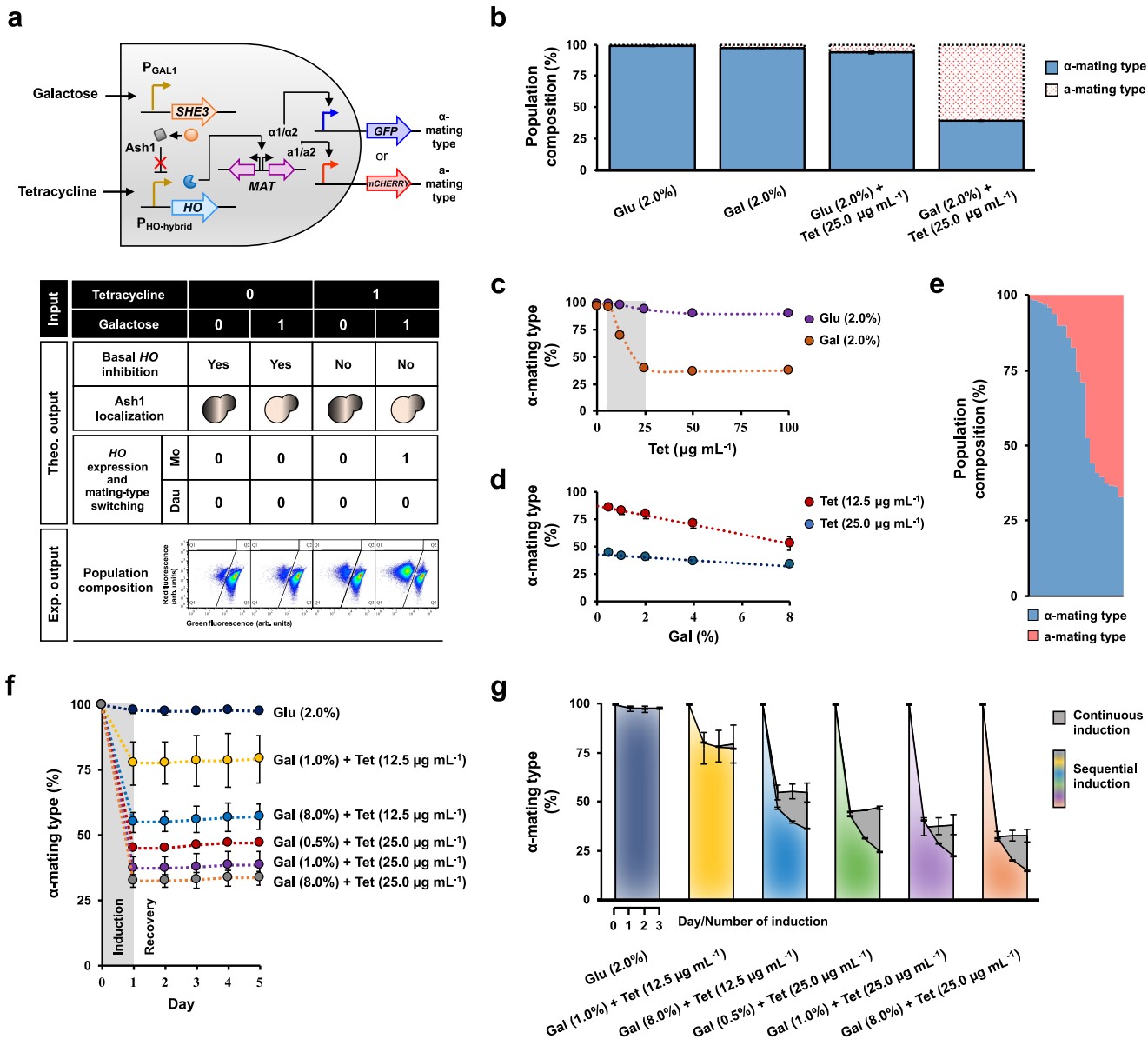

**Fig. 3 | An AND logic gate for tunable synthetic microbial consortium composition control. a** Substitution of the leaky *HO* promoter with a hybrid promoter (P$_{HO-hybrid}$) resulted in an AND logic gate co-modulated by both galactose and tetracycline. Basal *HO* inhibition was established by constitutive expression of TetR (from the P$_{TEF1}$-*TetR*-T$_{CYC1}$ module) to block the *tetO* sites and derepressed by the addition of tetracycline. **b** The AND logic gate tightened asymmetric *HO* expression and mating-type switching, forming a microbial consortium only in the presence of both inducers. $n = 6$ biological replicates. The AND logic gate exhibited dose-dependent responses to different concentrations of **c** tetracycline and **d** galactose.

$n = 6$ biological replicates. **e** The diversity of population composition formed by the AND logic gate. **f** The stability of the microbial consortia formed by the AND logic gate in a medium without inducers. $n = 6$ biological replicates. **g** Changes in the population composition of the microbial consortia formed by the AND logic gate following continuous inductions ($n = 6$ biological replicates) in the same inducing medium or sequential inductions ($n = 3$ biological replicates) in a fresh inducing medium. Values shown in panels **b**–**g** represent the mean ± standard deviation. Mo mother cell, dau daughter cell, glu glucose, gal galactose, tet tetracycline. Source data are provided as a Source Data file.

diluted (20-fold) and passaged the consortia through two additional rounds in a fresh-inducing medium and monitored changes in their population compositions. Figure 3g reveals that all five consortia experienced a progressive decline in the proportion of α-mating type haploids with each subsequent induction. This indicates that reinduction is viable, provided inducers are replenished, and the cell population is not saturated. However, it is also noteworthy that the rate of decrease in the proportion of α-mating type haploids during successive inductions was lower than in the initial induction. The reason behind this trend remains unclear but is likely related to the limited availability of α-mating type haploids eligible for switching.

## Biotransformation through division of role and microbial consortia cooperativity

We subsequently explored the utility of the AND logic gate in facilitating cooperativity within microbial consortia for metabolic engineering applications. The conceptual framework posits that cooperativity can be achieved by segregating the expression responsibilities of a biosynthetic pathway across two haploids of opposite mating types, which sequentially convert a substrate into a desired product (Fig. 4a). This approach necessitates dividing the pathway into two segments and controlling the expression of the enzymes for each segment with mating type-specific promoters. As a demonstration of the concept, the xylan degradation pathway was selected. Previous

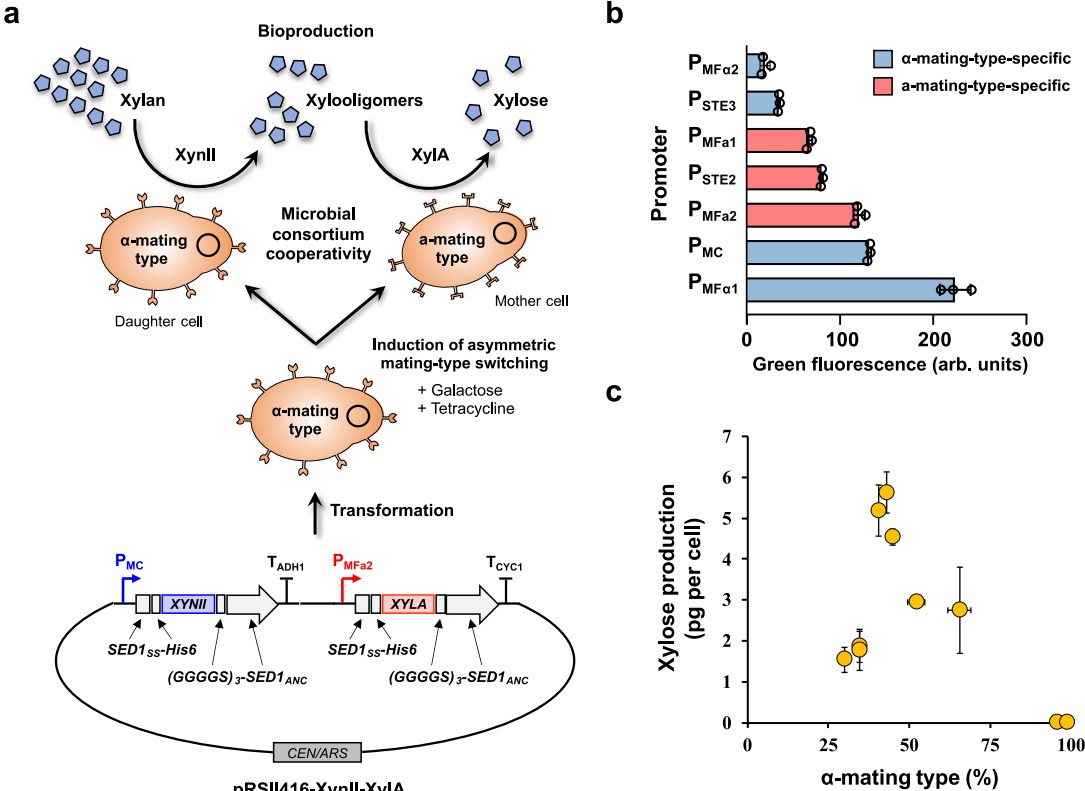

**Fig. 4 | Implementation of microbial consortium cooperativity for xylose bioproduction. a** Plasmid pRSII416-XynII-XylA was constructed for differential cell-surface expression of *T. reesei* xylanase, XynII, and *A. oryzae* β-xylosidase, XylA. Cell surface expression was achieved by fusing the two enzymes with the secretion signal (Sed1$_{SS}$) and the anchor domain (Sed1$_{ANC}$) of the yeast cell wall protein Sed1 at the N- and C-terminal, gapped by a 6× histidine-tag (His6) before and a 3× glycine-glycine-glycine-glycine-serine (GGGGS)$_3$ flexible linker after. The substrate xylan was first converted to xylooligomers by XynII expressed from the α-mating-type-specific promoter, P$_{MC}$, in the α-mating type haploids, and subsequently to xylose by XylA expressed from the a-mating-type-specific promoter, P$_{MFa2}$, in the a-mating type haploids. **b** Characterization and comparison of the expression strength of four α-mating-type-specific promoters (P$_{MFα1}$, P$_{MC}$, P$_{MFα2}$, and P$_{STE3}$), and three a-mating-type-specific promoters (P$_{MFa1}$, P$_{MFa2}$, and P$_{STE2}$). $n = 3$ biological replicates. **c** Xylose bioproduction per cell from xylan using microbial consortia formed by inducing the transformant strain, MTS019 pRSII416-XynII-XylA, with different concentrations of tetracycline (12.5 and 25.0 µg mL$^{-1}$) and galactose (0.5, 1.0 and 8.0%). $n = 3$ biological replicates. Values shown in panels (b) and (c) were the mean ± standard deviation. Source data are provided as a Source Data file.

studies have demonstrated that engineered yeast strains co-displaying on their cell surface *Trichoderma reesei*'s xylanase II (XynII; EC 3.2.1.8) and *Aspergillus oryzae*'s β-xylosidase (XylA; EC 3.2.1.37), can degrade extracellular xylan into xylooligosaccharides and subsequently into xylose[42–44]. Notably, pentose sugar, such as xylose cannot be readily utilized by *S. cerevisiae*[45], which eased the quantification of xylose bioproduction.

The plasmid pRSII416-XynII-XylA was constructed for differential expression of *XYNII* and *XYLA* genes. Cell surface display was achieved by fusing the two enzymes with the secretion signal (Sed1$_{SS}$) and the anchor domain (Sed1$_{ANC}$) of the yeast cell wall protein Sed1 at the N- and C-terminal, respectively[46–48]. Prior to this construction, we characterized three α-mating-type-specific promoters (P$_{MFα1}$, P$_{MFα2}$, and P$_{STE3}$) and three a-mating-type-specific promoters (P$_{MFa1}$, P$_{MFa2}$, and P$_{STE2}$) to identify a promoter pair with comparable expression strengths (Fig. 4b). However, none of the native promoters formed a perfectly matched pair, prompting the creation of a hybrid α-mating-type-specific promoters, P$_{MC}$ (Fig. 4b and S4). This hybrid promoter was created by replacing the TATA box-containing sequence downstream of the *MFα1* promoter (−275 to −1 bp) with that from the *CYC1* promoter (−252 to −1 bp), while keeping the upstream sequence (−1000 to −276 bp) containing the P'Q elements essential for α-mating-type-specific expression intact[49–52]. With an expression strength half that of the *MFα1* promoter, this hybrid promoter formed a matching pair with the a-mating-type-specific promoter, P$_{MFa2}$. The *MC* and *MFa2* promoters were utilized to regulate the expression of the *XYNII* and

*XYLA* genes, respectively, in the plasmid construct, resulting in *XYNII* expression in the α-mating type haploids and *XYLA* expression in the a-mating type haploids.

To validate xylose bioproduction via cooperativity within microbial consortia, we transformed the MTS019 strain with the plasmid pRSII416-XynII-XylA and subsequently induced the resulting transformant strain with different concentrations of tetracycline (12.5 and 25.0 µg mL$^{-1}$) and galactose (0.5%, 1.0%, and 8.0%). The consortia formed were subjected to biotransformation with 1% xylan for one day, after which the xylose concentration was quantified using high-performance liquid chromatography (HPLC). Contrary to the non-induced controls, which consisted of nearly homogeneous populations with at least 95% α-mating type haploids and did not produce xylose, all consortia exhibiting mating-type heterogeneity succeeded in producing xylose from xylan (Fig. 4c). Notably, higher xylose production per cell was observed in consortia with a balanced proportion of haploids of opposite mating types (40–60% α-mating type). Given the matched promoter strengths of P$_{MC}$ and P$_{MFa2}$, this indicates that the XynII and XylA enzymes are equally efficient, necessitating a balanced presence for optimal productivity. Should one enzyme exhibit lower activity than the other, an increased quantity of the less active enzyme and its corresponding mating type would be necessary. This offers versatile tools for tuning and determining the ratio of pathway enzymes to achieve optimum flux.

To compare the performance of microbial consortia with the typical single-strain monoculture system in xylose bioproduction, we

constructed both the α- and a-mating type derivatives of strain MTS019, which were unable to switch mating types. This was accomplished by replacing the dominant *HO* gene in strain MTS019 with the recessive *ho* gene. The resulting strains, MTS031α (α-mating type) and MTS031a (a-mating type) were transformed with plasmid derivatives of pRSII416-XynII-XylA. These plasmids, denoted as pRSII416-MCp-XX and pRSII416-MFa2p-XX, utilized either the *MC* or *MFa2* promoter to regulate the expression of both the *XYNII* and *XYLA* genes (Figs. S6a and S6b). This design rendered *XYNII* and *XYLA* expression in the α-mating type MTS031α strain for the former and in the a-mating type MTS031a strain for the latter. Strains MTS019, MTS031α, MTS031a, and their respective plasmid transformants exhibited comparable growth rates, with mean doubling times ranging from approximately 81.5 to 89.8 min (one-way ANOVA, *P* value = 0.7860). This indicated that there were no growth differences due to mating type or differential expression of *XYNII* and *XYLA* genes (Fig. S7).

As shown in Fig. S6c, xylose production per cell by the MTS031α pRSII416-MCp-XX and MTS031a pRSII416-MFa2p-XX strains were ~4-fold higher than the maximum observed in the microbial consortia established by strain MTS019 pRSII416-XynII-XylA. Although the cell surface display of XynII and XylA enzymes bypass the need for membrane crossing for intermediate exchange, the proximity of these enzymes on the same cell likely explains the higher productivity of the monocultures. Nonetheless, we believe that our system will be advantageous in scenarios where the expression of a large biosynthetic pathway in a single chassis is limiting, making cooperativity within a microbial consortium beneficial. The differential expression of such biosynthetic pathway would be greatly facilitated by leveraging the collection of mating-type-specific promoters characterized in this study.

## Discussion

In this study, we present a synthetic biology approach that repurposes the yeast mating-type switching mechanism into a genetic logic gate capable of inducing cell differentiation, generating consortia with mating-type heterogeneity from an initially homogeneous population. By modulating the concentrations of the inducers, we established a method for controlling population composition by differentially regulating the asymmetric *HO* expression and mating-type switching processes. Additionally, we demonstrated the utility of the consortia formed in facilitating cooperativity, particularly between two isogenic haploids of opposite mating types, each expressing a unique xylanolytic enzyme, working sequentially to convert xylan into xylose.

We posit that the approach delineated in this study represents a significant advancement in the development of synthetic microbial consortia. Our approach offers the advantage of utilizing isogenic consortium members derived from the same strain, thereby enhancing stability as the members can be readily differentiated from one another. Indeed, instability arising from differential cell growth, incompatibility, and competition has been a major issue hampering synthetic microbial consortia. A recent study addressed this challenge by fostering interdependency among consortium members[53]. Specifically, the authors engineered and deployed cross-feeding auxotrophic and overexpression yeast strains to construct two- and three-member consortia and demonstrated their stability and effectiveness in enhancing resveratrol production through division of labor. While our present study focused on constructing a two-member consortium, our approach holds potential for further adaptation. We envisage the incorporation of additional homothallic yeast strains and their combinations to expand the genetic diversity, complexity, and functionality of the microbial consortia formed. Moreover, the inducibility of our approach eliminates the need for manual mixing of various consortium members, enabling microbial consortia formation and cooperativity in environments where direct human intervention is impractical. Future iterations could explore using alternative inducible

promoters to regulate *SHE3* expression to facilitate the formation of microbial consortia and cooperativity under specific environmental conditions.

The ability to generate microbial consortia with diverse population compositions holds significant promise in synthetic biology, particularly for optimizing bioproduction. By distributing biosynthetic enzymes into two parts expressed under haploids of opposite mating types, the optimal enzyme ratio can be inferred by observing the population composition needed for maximum productivity. Additionally, apart from modulating the population composition, the ratios of these enzymes could be adjusted using the seven mating-type-specific promoters characterized in this study. While their varying expression strengths offer advantages in fine-tuning enzyme expression, the moderate expression strengths of these mating-type specific promoters may pose limitations in certain contexts. Further investigation is therefore warranted to explore engineering strategies that can enhance the expression strengths of these promoters while preserving their mating-type specificity. In summary, our study offers a versatile toolkit for the generation and precision-tuning of functionally heterogeneous yet isogenic yeast consortia. This contributes to the advancement of microbial consortia cooperativity and opens additional avenues for biotechnological applications.

## Methods
### Strains and culture conditions
*S. cerevisiae* haploid BY4741, haploid BY4742, and diploid BY4743 were purchased from the American Type Culture Collection (ATCC). For non-selective cultivation, yeast strains were grown in YPD medium (1% yeast extract, 2% peptone, and 2% glucose) or synthetic defined medium (6.7 g L$^{-1}$ yeast nitrogen base, 1.92 g L$^{-1}$ yeast synthetic drop-out medium supplements without uracil, 85.6 mg L$^{-1}$ uracil and 2% glucose or 2% galactose). Yeast prototrophs (His$^+$, Lys$^+$, Met$^+$, or Ura$^+$) were grown in a synthetic minimal medium lacking the corresponding essential nutrient (6.7 g L$^{-1}$ yeast nitrogen base, 2% glucose and an appropriate yeast synthetic drop-out medium supplements). For the selection of *URA3* marker loss, yeast strains were grown in a synthetic defined medium supplemented with 1 mg mL$^{-1}$ of 5-fluoroorotic acid (5-FOA). Solid culture media were similarly prepared with the addition of 2.5% agar. All yeast strains were grown at 30 °C.

### Plasmid and strain construction
Genome-edited yeast strains were constructed using the "pop in/pop out" gene replacement strategy[37,54]. For this, a collection of locus-specific integration plasmids was first generated using plasmid pYES2/INTG as the vector backbone (Table 1). Plasmid pYES2/INTG was a truncated version of the plasmid pYES2/CT (Thermo Fisher Scientific) that lacked sequences between the *2μ* origin and the *GAL1* promoter. Integration plasmids for the deletion of *MFα1-2*, *STE3*, and *SHE1-3* genes were constructed by cloning 500–1000 bp upstream and downstream flanking sequences of the target gene into the multiple cloning site (MCS) of the pYES2/INTG vector. Integration plasmids for DNA substitution were similarly constructed by cloning *GFP*, *mCHERRY*, *HIS3*, *HO*, *ho*, P$_{GAL1}$-*HO*-T$_{CYC1}$, P$_{GAL1}$-*GFP*-T$_{CYC1}$, P$_{GAL1}$-*SHE3*-T$_{CYC1}$, P$_{TEF1}$-*TetR*-T$_{CYC1}$, P$_{HO-hybrid}$ and P$_{MC}$ in between the flanking sequences (500-1000 bp) of the target genome site. The DNA sequences encoding *GFP*, *mCHERRY*, *TetR* (*E. coli*; UniProtKB: P04483), and the promoter sequence of P$_{TX}$[38] were synthesized by Integrated DNA Technologies (Singapore) and codon-optimized for expression in yeast where applicable. DNA sequences native to *S. cerevisiae* were PCR-amplified directly from the genomic DNAs. The dominant *HO* gene was obtained by mutating the recessive *ho* gene (G565A, A667G, C1214T, and T1424A[55,56]) through splicing by overlap-extension PCR using customized mutagenic primers. Promoter P$_{MC}$ was constructed by combining the −1000 to −276 bp sequence upstream of the *MFα1* gene[49–51] with the 252 to −1 bp sequence upstream of the *CYC1* gene[52].

**Table 1 | List of plasmids used in this study**

| Plasmid | Genotype | Source |
|---|---|---|
| pYES2/CT | $P_{GAL1}$-MCS-$T_{CYC1}$; *URA3*; *2μ ori*; *AmpR*; *ori*; *f1 ori*. | Thermo FS |
| pYES2/INTG | Modified from pYES2/CT by removing sequences between *2μ ori* and $P_{GAL1}$. | This study |
| pRSII416 | MCS; *URA3*; *CEN/ARS*; *AmpR*; *ori*; *f1 ori*. | Addgene |
| pRSII416-XynII-XylA | pRSII416 with $P_{MC}$-*SED1$_{SS}$*-*His6*-*XYNII*-*(GGGGS)$_3$*-*SED1$_{ANC}$*-$T_{ADH1}$ and $P_{MFa2}$-*SED1$_{SS}$*-*His6*-*XYLA*-*(GGGGS)$_3$*-*SED1$_{ANC}$*-$T_{CYC1}$ inserted into the MCS. | This study |
| pRSII416-MCp-XX | pRSII416 with $P_{MC}$-*SED1$_{SS}$*-*His6*-*XYNII*-*(GGGGS)$_3$*-*SED1$_{ANC}$*-$T_{ADH1}$ and $P_{MC}$-*SED1$_{SS}$*-*His6*-*XYLA*-*(GGGGS)$_3$*-*SED1$_{ANC}$*-$T_{CYC1}$ inserted into the MCS. | This study |
| pRSII416-MFa2p-XX | pRSII416 with $P_{MFa2}$-*SED1$_{SS}$*-*His6*-*XYNII*-*(GGGGS)$_3$*-*SED1$_{ANC}$*-$T_{ADH1}$ and $P_{MFa2}$-*SED1$_{SS}$*-*His6*-*XYLA*-*(GGGGS)$_3$*-*SED1$_{ANC}$*-$T_{CYC1}$ inserted into the MCS. | This study |

All integration plasmids were constructed using the standard restriction digestion and ligation method, validated by Sanger sequencing (1st Base, Singapore), and harvested from *E. coli* TOP10 cells grown in lysogeny broth (LB) added with $100\,\mu g\,mL^{-1}$ ampicillin. Upon construction, $0.5–1\,\mu g$ of the integration plasmids were linearized at either flanking sequence by restriction digestion and subsequently transformed into appropriate yeast strains using the standard lithium acetate/single-stranded carrier DNA/polyethylene glycol method[57]. Positive integrant colonies were selected on a synthetic minimal medium lacking uracil for the presence of the *URA3* marker gene, which was part of the linearized integration plasmid construct. Marker rescue was performed by growing the positive integrant colonies overnight in YPD medium before selection on 5-FOA plates. The genomic modifications introduced were finally validated by colony PCR using specific primers. A list of genome-edited yeast strains constructed in this study is provided in Table 2.

Plasmid pRSII416-XynII-XylA (Table 1) for xylose bioproduction was constructed through the stepwise cloning of the DNA fragments constituting the $P_{MC}$-*XYNII*-$T_{ADH1}$ and $P_{MFa2}$-*XYLA*-$T_{CYC1}$ expression modules into the MCS of the pRSII416 vector (Addgene) through a standard restriction digestion and ligation method. The DNA sequences encoding genes *XYNII* (*T. reesei*; GenBank: ACB38137.1; 33–222 aa[42]) and *XYLA* (*A. oryzae*; GenBank: BAA28267.1; 21–798 aa[43,44]) were synthesized by Integrated DNA Technologies (Singapore) and codon-optimized for expression in yeast. Both genes were fused with the secretion signal (*SED1$_{SS}$*; 1–57 bp[46]) and the anchor domain (*SED1$_{ANC}$*; 328–1017 bp[47,48]) of the *SED1* gene, gapped by DNA sequences encoding a 6× histidine-tag (His6) before and a 3× glycine-glycine-glycine-glycine-serine (GGGGS)$_3$ flexible linker after. Plasmids pRSII416-MCp-XX and pRSII416-MFa2p-XX were constructed by replacing the $P_{MFa2}$-*XYLA*-$T_{CYC1}$ and $P_{MC}$-*XYNII*-$T_{ADH1}$ of plasmid pRSII416-XynII-XylA, with $P_{MC}$-*XYLA*-$T_{CYC1}$ and $P_{MFa2}$-*XYNII*-$T_{ADH1}$, respectively (Figs. S6a and S6b). The plasmids were similarly transformed into yeast strains using the standard lithium acetate/single-stranded carrier DNA/polyethylene glycol method[57]. Positive transformant colonies were selected on a synthetic minimal medium lacking uracil as a uracil prototroph. The sequences of the primers and plasmids generated in this study are provided in a Supplementary Data file.

### Spotting assays

To examine the impact of *MFα1-2* and *STE3* gene deletion on mating efficiency, $5 \times 10^6$ cells of each of the strains MTS001–MTS007 (and the control BY4742) were mixed with equivalent cells of the haploid BY4741 in 1 mL YPD medium. The resulting mating pairs were incubated at 30 °C for 4.5 h without shaking to facilitate conjugation. A diploid control, BY4743, was also included in this assay, in which $10^7$ cells were incubated. For the semi-quantitative genetic assay to examine the impact of *SHE1-3* gene deletion on *HO* expression, strains MTS011–MTS017 (and the control MTS010) harboring a $P_{HO}$-*HIS3*-$T_{HO}$ reporter module were incubated with an initial $OD_{600}$ of 0.6 in the synthetic defined medium for 6 hours, at 30 °C and 225 rpm. Prior to spotting onto an appropriate selection plate (lysine and methionine

drop-out for the mating assay, and histidine drop-out for the genetic assay) at a volume of $5\,\mu L$, all samples were washed three times, resuspended to a final $OD_{600}$ of 10, and serially diluted (10-fold) using deionized water. Images of the selection plate were taken after 2 days of incubation at 30 °C using an Amersham Imager 600 (GE Healthcare Life Sciences).

### Growth curve characterization

Overnight seed cultures of strains BY4742, MTS019, MTS031α, MTS031a, MTS019 pRSII416-XynII-XylA, MTS031α pRSII416, MTS031α pRSII416-MCp-XX, and MTS031a pRSII416-MFa2p-XX were resuspended in synthetic defined medium or synthetic minimal medium lacking uracil containing 2% glucose to an initial $OD_{600}$ of 0.1. The samples were then loaded at a volume of $100\,\mu L$ onto a 96-well microplate and subjected to a 20-hour kinetic analysis using a Synergy H1 Multi-Mode Reader (BioTek Instruments). Cells were incubated at 30 °C, 807 cpm, with $OD_{600}$ measured at a 15-min interval. Doubling time was calculated by dividing Ln(2) by the exponent of cell density versus time during the exponential phase (135–660 min). The mean doubling time was determined by averaging the doubling times of the three biological replicates.

### Fluorescence microscopy

All microscopic images were taken at ×100 magnification using a Leica DMi8 Inverted Microscope and processed using Leica Application Suite X software (Leica Microsystems).

### Flow cytometry

To characterize the expression strength of mating-type-specific promoters, overnight seed cultures of the *GFP*-integrated strains, MTS020-MTS025, MTS026α, MTS026a, MTS027α, and MTS027a, were resuspended to an initial $OD_{600}$ of 0.2 in synthetic defined medium containing 2% glucose. The starting cultures were loaded at a volume of $100\,\mu L$ onto a 96-well microplate and cultivated at 30 °C, 999 rpm for 24 h before flow cytometry measurement. Strains MTS028-MTS030 (and the control BY4742) were similarly prepared and incubated in a synthetic defined medium containing 2% glucose for the examination of leaky *GFP* expression from the *HO* promoter. Fluorescence scatter plots of strains BY4742, MTS008α, MTS008a, MTS027α, MTS027a, and MTS030 were generated using cells similarly prepared, except that the MTS030 strain was incubated in a synthetic defined medium containing 2% galactose.

For the characterization of the two YES and the AND logic gates, starting cultures of strains MTS009, MTS018, and MTS019 were similarly prepared but treated with different combinations of glucose (0–2%), galactose (0–8%) and tetracycline (0–100 μg mL⁻¹). Cells were cultivated at a volume of 100 μL on a 96-well microplate, at 30 °C, 999 rpm. The cultures, grown after 24 h of treatment (1 day of treatment), were subjected to two rounds of daily passage (5 μL of culture diluted) in 100 μL of synthetic defined medium containing 2% glucose (2 days of recovery) before flow cytometry measurement.

**Table 2 | List of strains used in this study**

| Strain | Genotype | Source |
|--------|----------|--------|
| BY4741 | MATa ho HMRa^stuck his3Δ1 leu2Δ0 met15Δ0 ura3Δ0 | ATCC |
| BY4742 | MATα ho HMRa^stuck his3Δ1 leu2Δ0 lys2Δ0 ura3Δ0 | ATCC |
| BY4743 | MATa/α ho/ho HMRa^stuck/HMRa^stuck his3Δ1/his3Δ1 leu2Δ0/leu2Δ0 LYS2/lys2Δ0 met15Δ0/MET15 ura3Δ0/ura3Δ0 | ATCC |
| MTS001 | BY4742 MATα mfα1Δ | This study |
| MTS002 | BY4742 MATα mfα2Δ | This study |
| MTS003 | BY4742 MATα ste3Δ | This study |
| MTS004 | BY4742 MATα mfα1Δ mfa2Δ | This study |
| MTS005 | BY4742 MATα mfα1Δ ste3Δ | This study |
| MTS006 | BY4742 MATα mfα2Δ ste3Δ | This study |
| MTS007 | BY4742 MATα mfα1Δ mfα2Δ ste3Δ | This study |
| MTS008α | BY4742 MATα mfα1Δ::mCHERRY mfα1Δ::GFP mfα2Δ ste3Δ | This study |
| MTS008a | BY4742 MATa mfα1Δ::mCHERRY mfα1Δ::GFP mfα2Δ ste3Δ | This study |
| MTS009 | BY4742 MATm hoΔ::Inv(P_{GAL1}-HO-T_{CYC1}) mfα1Δ::mCHERRY mfα1Δ::GFP mfα2Δ ste3Δ | This study |
| MTS010 | BY4742 MATα hoΔ::HIS3 mfα1Δ mfα2Δ ste3Δ | This study |
| MTS011 | BY4742 MATα hoΔ::HIS3 mfα1Δ mfα2Δ ste3Δ she1Δ | This study |
| MTS012 | BY4742 MATα hoΔ::HIS3 mfα1Δ mfα2Δ ste3Δ she2Δ | This study |
| MTS013 | BY4742 MATα hoΔ::HIS3 mfα1Δ mfα2Δ ste3Δ she3Δ | This study |
| MTS014 | BY4742 MATα hoΔ::HIS3 mfα1Δ mfα2Δ ste3Δ she1Δ she2Δ | This study |
| MTS015 | BY4742 MATα hoΔ::HIS3 mfα1Δ mfα2Δ ste3Δ she1Δ she3Δ | This study |
| MTS016 | BY4742 MATα hoΔ::HIS3 mfα1Δ mfα2Δ ste3Δ she2Δ she3Δ | This study |
| MTS017 | BY4742 MATα hoΔ::HIS3 mfα1Δ mfα2Δ ste3Δ she1Δ she2Δ she3Δ | This study |
| MTS018 | BY4742 MATm hoΔ::HO mfα1Δ::mCHERRY mfα1Δ::GFP mfα2Δ ste3Δ she3Δ::Inv(P_{GAL1}-SHE3-T_{CYC1}) | This study |
| MTS019 | BY4742 MATm p_{ho}-hoΔ::P_{HO-hybrid}-HO mfα1Δ::mCHERRY mfα1Δ::GFP mfα2Δ::Inv(P_{TEF1}-TetR-T_{CYC1}) ste3Δ she3Δ::Inv(P_{GAL1}-SHE3-T_{CYC1}) | This study |
| MTS020 | BY4742 MATα mfα1Δ::GFP | This study |
| MTS021 | BY4742 MATα mfα2Δ::GFP | This study |
| MTS022 | BY4742 MATα ste3Δ::GFP | This study |
| MTS023 | BY4741 MATa mfa1Δ::GFP | This study |
| MTS024 | BY4741 MATa mfa2Δ::GFP | This study |
| MTS025 | BY4741 MATa ste2Δ::GFP | This study |
| MTS026α | BY4742 MATα mfα2Δ ste3Δ p_{mfα1}-mfα1Δ::P_{MC}-GFP | This study |
| MTS026a | BY4742 MATa mfα2Δ ste3Δ p_{mfα1}-mfα1Δ::P_{MC}-GFP | This study |
| MTS027α | BY4742 MATα mfα1Δ::GFP mfα2Δ ste3Δ | This study |
| MTS027a | BY4742 MATa mfα1Δ::GFP mfα2Δ ste3Δ | This study |
| MTS028 | BY4742 MATα hoΔ::GFP mfα1Δ mfα2Δ ste3Δ | This study |
| MTS029 | BY4742 MATα hoΔ::GFP mfα1Δ mfα2Δ ste3Δ she3Δ | This study |
| MTS030 | BY4742 MATα mfα1Δ mfα2Δ ste3Δ she3Δ::Inv(P_{GAL1}-GFP-T_{CYC1}) | This study |
| MTS031α | BY4742 MATα p_{ho}-hoΔ::P_{HO-hybrid}-ho mfα1Δ::mCHERRY mfα1Δ::GFP mfα2Δ::Inv(P_{TEF1}-TetR-T_{CYC1}) ste3Δ she3Δ::Inv(P_{GAL1}-SHE3-T_{CYC1}) | This study |
| MTS031a | BY4742 MATa p_{ho}-hoΔ::P_{HO-hybrid}-ho mfα1Δ::mCHERRY mfα1Δ::GFP mfα2Δ::Inv(P_{TEF1}-TetR-T_{CYC1}) ste3Δ she3Δ::Inv(P_{GAL1}-SHE3-T_{CYC1}) | This study |

Inv, insertion in an inverted orientation relative to the gene; MATm, mixture of haploids of opposite mating types.

This allowed sufficient time for the degradation of fluorescent proteins produced by the previous mating type in the absence of inducers. The induction timing was set to 24 h because, by this duration, the cells would reach saturation (Fig. S7). To examine the stability of the population generated by the AND logic gate, cultures of strain MTS019 formed after 1 day of treatment and 2 days of recovery were further passaged for an additional four days in a synthetic defined medium containing 2% glucose. For the sequential induction assays, cultures of strain MTS019 were subjected to three rounds of the 1-day treatment plus 2-day of recovery procedures. For the continuous induction assays, cultures of strain MTS019 were incubated in the same medium for an additional two days, with daily samplings similarly done by first passaging the grown cultures two rounds in a synthetic defined medium containing 2% glucose before flow cytometry measurement.

All samples were cultivated in a 96-well microplate using a Multitron Pro microplate shaker (Infors HT). All samples were diluted appropriately with deionized water prior to fluorescence measurement using a BD Accuri™ C6 flow cytometer equipped with a blue laser (488 nm) and a BD CSampler™. Typically, 100,000 cells were analyzed at a flow rate of $35\,\mu L\,min^{-1}$ (core size 16 μm) to estimate the mean fluorescence or the population composition of a sample. Green fluorescence and red fluorescence were detected by the FL1 (520/30 nm) and FL4 (610/20 nm) channels, respectively. All data acquired were analyzed using FlowJo v10.10 software (BD Biosciences).

**Xylose bioproduction**
Starting cultures for xylose bioproduction using different microbial consortia were similarly prepared by growing strain MTS019 transformed with the pRSII416-XynII-XylA plasmid in synthetic minimal medium lacking uracil treated with different combinations of glucose (0–2%), galactose (0–8%) and tetracycline (0–100 μg mL⁻¹). The cultures formed after 1 day of treatment and 2 days of recovery were inoculated to an initial $OD_{600}$ of 0.2 in synthetic minimal medium lacking uracil containing 2% glucose and grown with 1% beechwood xylan (Megazyme) at 30 °C, 225 rpm for 24 h.

For xylose bioproduction using a single-strain monoculture system, overnight cultures of transformant strains MTS031α pRSII416-MCp-XX and MTS031a pRSII416-MFa2p-XX, were similarly inoculated to an initial $OD_{600}$ of 0.2 in synthetic minimal medium lacking uracil containing 2% glucose and incubated with 1% beechwood xylan (Megazyme) at 30 °C, 225 rpm for 24 h. Strains MTS031α pRSII416, MTS031α pRSII416-MFa2p-XX, MTS031a pRSII416 and MTS031a pRSII416-MCp-XX were included as controls.

Aliquots of the resulting cultures were subjected to flow cytometry measurements to assess cell density and population composition. Cell density was calculated based on the volume of culture required to capture 100,000 cells, at a flow rate of $35\,\mu L\,min^{-1}$ (core size 16 μm) using the flow cytometer. The rest of the cultures were centrifuged at $21,000 \times g$ for 10 min to obtain supernatants for the quantification of xylose concentration using an Agilent 1260 Infinity LC system (Agilent Technologies) equipped with an Aminex HPX-87H column (Bio-Rad). Separation was performed using 5 mM sulfuric acid as the mobile phase at a flow rate of 0.6 mL min⁻¹ and a column temperature of 55 °C. Compounds were detected using a refractive index detector with an optical unit temperature of 55 °C. Representative chromatograms were provided in Fig. S5.

**Statistical analysis and reproducibility**
All experiments were conducted with at least three biological replicates. Statistical significance was inferred by conducting either a one-

way analysis of variance (ANOVA) or a two-tailed Student's *t*-test with 95% confidence intervals.

## Reporting summary

Further information on research design is available in the Nature Portfolio Reporting Summary linked to this article.

## Data availability

All data generated in this study are provided in the Source Data file. Source data are provided with this paper.

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

## Acknowledgements

This work was supported by NUS Medicine Synthetic Biology Translational Research Program (NUHSRO/2024/064/NUSMed/05/SynCTI2.0, M.W.C.), Investigatorship (NRF-NRFI05-2019-0004, M.W.C.) and Competitive Research Program (NRF-CRP27-2021-0004, M.W.C.) of the National Research Foundation of Singapore, the Industry Alignment Fund (I2301E0021, M.W.C.), and the National Centre for Engineering Biology (NCEB) (NRF-MSG-2023-0003, M.W.C.). This work used the resources of the Singapore BioFoundry, a bio-manufacturing research facility located at the National University of Singapore.

## Author contributions

Y.C.H., J.L.F., and M.W.C. conceived and designed the study. Y.C.H., S.K., and A.V.S. performed the study. Y.C.H., J.L.F., and M.W.C. analyzed the data. Y.C.H., J.L.F., and M.W.C. wrote and edited the manuscript. M.W.C. obtained funding and supervised the study. All authors critically reviewed the manuscript and approved the final version of the manuscript for submission.

## Competing interests

The authors declare no competing interests.
