## [Peer Review File · Nature Communications]

Reviewers' Comments:

Reviewer #1:

Remarks to the Author:

This manuscript describes a synthetic biology approach to switch mating-types in yeast and adjust the ratio of a-type and alpha-type. The determination of mating-type in yeast is inherently controlled by a complex and precise system, but ultimately a ratio control system was constructed by constructing an AND logic gate, an artificial gene expression circuit. In this study, careful experiments are being carried out to suppress leaks in gene expression. The final yeast form shown in Figure 3 shows that fine tuning of the consortium can be achieved by changing the amounts and ratios of the two types of expression inducers. In addition, the authors succeeded in artificially generating diverse yeast consortia with different mating-type ratios, and demonstrated the effectiveness of this method in the production of xylose from xylan polymer. This study yielded important results that demonstrate the potential of yeast synthetic biology while there are also inadequacies in the interpretation of the results obtained and in the data collected.

1. In Figure 1D, two types of populations are observed with α -Mating type flow cytometry, but it should be explained what the left side represents (the gray population in Mixture).
2. In the legend of Figure 2A, the logic gate table should be explained, such as what Dau and Mo represent.
3. Although the stability of the microbial consortiums mating-type composition is highlighted in Figure 3F, stability may be affected by the cell density at that time. The authors should consider the aspect of cell density and, if necessary, conduct experiments using cells with different densities.
4. Figure 4C shows the relationship between α -Mating Type % and xylose concentration, but this data alone is insufficient to understand the phenomenon. If the amount of xylose produced is explained only by the ratio of mating-types, there is a risk that the understanding of what is happening may become unclear. In order to understand what is actually happening, the authors should also show the number of cells per solution for each yeast, in addition to the α -type Xylanase activity and a-type xylosidase activity. The ability of *Saccharomyces cerevisiae* to assimilate xylose must also be taken into consideration.
5. It would be easier for readers to understand if the name of the gene to be deleted was specifically written on line 112.
6. In lines 142 to 144, the authors claim that modulating HO expression is typically unfeasible, but why not also state the reason?
7. In lines 215 to 216, the authors claim leaky HIS3 expression but the expression data is missing. The authors should be better to analyze transcript of HIS3 gene.

8. In lines 267-268, the authors mention that the varied responses to different inducer concentrations produced a diversity of population composition. However, the diversity would be dependent on the timing of induction by galactose and tetracycline. Therefore, the authors should explain how they decided on the induction timing.

9. In lines 279-280, the authors speculate that this stability could result from the depletion of one or both inducers, but how about directly quantifying the concentration of the inducer?

10. Line 282 mentions cell division, but the authors should collect data on the number of cells. In the first place, it is dangerous to discuss cell density based solely on OD, as cell size may have changed due to genetic manipulation.

11. When considering the application of the mating-type switching technology developed in this study, it is necessary to evaluate the cell proliferation ability of the constructed strain. Although this study collected data on the ratio of α -type and a-type, it would be better to also investigate the growth ability and doubling time of individual strains.

12. On line 353, the authors claim that determining the optimal enzyme ratios, but they should explain this application example more specifically. Will the technique of this study be an effective approach when optimizing the ratio of what enzymes?

Reviewer #2:

Remarks to the Author:

In this manuscript, entitled “Tunable cell differentiation via reprogrammed mating-type switching”, the authors present a well-written work with clever rational design features and a clear logic to it. By installing logic gates, they achieve a high degree of control over inducible mating-type switching in *S. cerevisiae*, which they then apply to division of roles in relation to biotechnology. Indeed, the topic is highly relevant and contributes to an active field of research. However, several minor and major issues need to be resolved prior to publication:

Minor comments:

1. Resolution appears to be low for flow cytometry plots. The sizes of flow cytometry plots are too small in 1D and 2A&E, and 3A. New subpopulations (plural) relative to Fig. 1D occur in the MAT α gate in 2A, 2E, and 3A, and, presumably due to poor reporter-performance of the MF α 1 promoter driving GFP, fluorescence profiles overlap – which is particularly obvious in the subpanel of Fig. 1D, where the subpopulation of this reporter is in Q4, and not in Q3 as otherwise would have been expected. Hence, I currently disagree with the line 656 statement “Expression of GFP and mCHERRY was mutually exclusive”.

It is unclear to me how the authors considered the MAT α sub-populations in the handling of their flow cytometry analyses throughout – particularly the ones that obviously float into the MAT α gate (Fig. 1D “mixture”). Please clarify in detail how you analyzed your flow cytometry data and explain how you took the overlapping subpopulations into account.

2. Line 131 and 133: Please change “GFP” and “mCHERRY” to italics as you are referring to genes here.

3. Line 49: Consider changing “(...) different cells (...)” to “(...) different cell types (...)”.

4. Fig. 2E and 3A do not properly illustrate the point that GFP or mCherry is expressed in MAT α or MAT α cells, respectively. Please consider changing the repression mark on mCherry to an arrowhead and mark the arrows going to GFP or mCherry by e.g., “ α -only” or “ α -only”, respectively.

5. Fig. 3G: “100” percent should be formatted to be in one line.

6. Please mention the severe limitation of strong promoter availability specific to the two mating types. It can be done in line 358 in connection with the claim “versatile toolkit”.

7. Please further discuss both advantages and disadvantages by your isogenic strain approach in relation to other recent demonstrations on division of labor in synthetic yeast co-cultures - e.g., Peng et al. (<https://www.nature.com/articles/s41564-023-01596-4>). This can be done in connection with lines 346-348 where, however, human intervention is not impractical.

8. I did not find the raw data e.g., for flow cytometry and LC-MS in the original submission.

Major comments:

1. References appear to be out of chronological order throughout the manuscript: Line 41: Reference 7 appears to be the first cited work. Line 79 to line 106: REFs 23+24 jump straight to REFs 41+42. Line 303 introduces REFs 25-27 (following REF 55 in line 244). Etc. Please make sure to correct it.

2. Line 162: Galactose-induction is binary, unless GAL2 is deleted (<https://doi.org/10.1074/jbc.M512317200>). The “titratable” experiments in Fig. 3D and

3G should be done again in a *gal2Δ* genetic background, particularly in the view of claiming “Further alterations in galactose concentration from 0.5 to 8.0% did not significantly change the population composition, which we conjectured reached equilibrium with the α-to-a mating type ratio maintained at approximately 0.14”. This experiment potentially makes the system more tunable.

3. Fig. 4B: Please show expression levels by GFP in both mating types for all promoters, and in particular for pMC, as it is a new hybrid promoter. This all talks into system control, and consequently the division of “role”.

4. Fig. 4C: It is necessary to show a “mono-culture” control for the conversion of xylan to xylose. Please benchmark to co-display on a strain that does not undergo mating-type switching (cited as REFs 25-27), by using strong constitutive promoters (TDH3 or CCW12).

Importantly, potential suboptimal performance in your system as compared to a mono-culture does not in my opinion prevent publication, but rather serves to help potential users of your technology to understand its advantages and limitations, and hence to plan their experimental work accordingly. Whatever the outcome, please include it in your discussion in the end.

Responses to reviewers' comments for manuscript submission NCOMMS-24-12398-T

We thank the reviewers for their comments. The point-by-point response to the comments is provided below.

Reviewer #1 (Remarks to the Author):

This manuscript describes a synthetic biology approach to switch mating-types in yeast and adjust the ratio of a-type and alpha-type. The determination of mating-type in yeast is inherently controlled by a complex and precise system, but ultimately a ratio control system was constructed by constructing an AND logic gate, an artificial gene expression circuit. In this study, careful experiments are being carried out to suppress leaks in gene expression. The final yeast form shown in Figure 3 shows that fine tuning of the consortium can be achieved by changing the amounts and ratios of the two types of expression inducers. In addition, the authors succeeded in artificially generating diverse yeast consortia with different mating-type ratios, and demonstrated the effectiveness of this method in the production of xylose from xylan polymer. This study yielded important results that demonstrate the potential of yeast synthetic biology while there are also inadequacies in the interpretation of the results obtained and in the data collected.

1. In Figure 1D, two types of populations are observed with α -Mating type flow cytometry, but it should be explained what the left side represents (the grey population in Mixture).

Thank you for the comment. The α -mating type shown in Figure 1D refers to strain MTS008 α , which was unable to switch mating type due to a recessive *ho* gene. Consequently, the population derived from this strain was homogeneously α -mating type. The subpopulation observed in Q4 of the flow cytometry plot for this strain (Figure S1A) represents α -mating type haploids with no green fluorescence (approximately 2.63% based on 144 independent samples). Their level of green fluorescence is notably comparable to the BY4742 control lacking any fluorescent reporter constructs (the grey population in Figure S1C). To improve clarity, we have now removed the BY4742 control from the "Mixture" flow cytometry plot in Figure 1D. Instead, the original plot that included the BY4742 control, is provided as supplementary information under Figure S1C.

We now added the following sentences to explain this in the manuscript.

Lines 131-142: "As illustrated in Figure 1D and S1, GFP was exclusively expressed in the α -mating type haploids (MTS008 α), and mCherry in the a-mating type haploids (MTS008a). Notably, red fluorescence higher than the BY4742 control but lower than the MTS008a strain was observed in the MTS008 α strain, which was later found to be caused by spillover signal from GFP detected by the red fluorescence channel (FL4). Both MTS008 α and MTS008a strains formed distinct populations with minimal overlap on a fluorescence scatter plot. However, direct quantification of population composition by evaluating the proportion of cells in Q3 and Q4 was challenging due to the presence of a small subpopulation of α -mating type haploids (approximately 2.63% based on 144 samples) that overlapped with the a-mating type haploids in Q4. To account for this, we estimated the proportion of α -mating type haploids in a population by dividing the percentage of cells in Q3 with 97.37% and determined the proportion of a-mating type haploids by subtracting the calculated percentage of α -mating type haploids from 100%."

Figure S1

2. In the legend of Figure 2A, the logic gate table should be explained, such as what Dau and Mo represent.

Thank you for the comment. We have now added the following sentences to the Figure 2A legend to explain the rationale of the YES logic gate and the abbreviations at the end of the legend.

Lines 792-794: “In this system, replacing the *HO* promoter eliminated the asymmetry between the haploid mother and daughter cells. *HO* expression and mating type switching in both cell types were activated by galactose and repressed by glucose.”

3. Although the stability of the microbial consortiums mating-type composition is highlighted in Figure 3F, stability may be affected by the cell density at that time. The authors should consider the aspect of cell density and, if necessary, conduct experiments using cells with different densities.

Thank you for the suggestion. As demonstrated by our subsequent experiments, high cell density affected the ability or the extent of the cells to switch mating type. However, this is unrelated to the performance of the AND logic gate but rather due to the suppression of cell division, which is a pre-requisite for mating-type switching to occur. As shown in Figure 3G, no significant changes in mating type composition occurred for cells continuously incubated in the same inducing medium

for up to 3 days. In contrast, re-passaging and diluting (20-fold) the same induced cells into fresh inducing medium resulted in changes in mating type composition.

In Figure 3F, the stability of the microbial consortium formed was assessed by monitoring changes in population composition after the removal of inducers. This was done by re-passaging and diluting (20-fold) the induced cells into fresh, non-inducing medium. The initial cell density was therefore low in this case, which did not limit the cells from switching. Thus, the maintenance of the unique mating type composition across five re-passages into fresh, non-inducing medium indicated the stability of the microbial consortium formed in the absence of inducers.

To improve clarity, we have now modified the sentences as follows:

Lines 292-297: “The stability of the microbial consortia formed was assessed by monitoring changes in population composition after the removal of inducers. As shown in Figure 3F, all five consortia, established after one day of induction in medium with varying inducer concentrations (12.5 or 25.0 $\mu\text{g mL}^{-1}$ tetracycline with 0.5, 1.0, or 8.0% galactose, or 2% glucose), consistently maintained their distinct population compositions for up to four days of repassaging (diluted 20-fold) in a fresh medium containing 2% glucose repressor. This indicated the stability of the consortia in the absence of inducers.”

4. Figure 4C shows the relationship between α -Mating Type % and xylose concentration, but this data alone is insufficient to understand the phenomenon. If the amount of xylose produced is explained only by the ratio of mating-types, there is a risk that the understanding of what is happening may become unclear. In order to understand what is actually happening, the authors should also show the number of cells per solution for each yeast, in addition to the α -type Xylanase activity and α -type xylosidase activity. The ability of *Saccharomyces cerevisiae* to assimilate xylose must also be taken into consideration.

Thank you for your suggestion. We collected data on the number of cells and replotted Figure 4C, showing the percentage of α -mating type versus xylose production per cell. This new plot accounts for the impacts of both the mating-type ratio and the number of cells on xylose production. We also note that *S. cerevisiae* strains, including BY4742, are known to not readily uptake and utilize pentose sugars including xylose (<https://www.nature.com/articles/srep19512>). We have now added the following sentences to discuss this point.

Lines 322-323: “Notably, pentose sugar such as xylose cannot be readily utilized by *S. cerevisiae*, which eased the quantification of xylose bioproduction.”

We have now added the following sentence to the Methods section to describe the method used to calculate cell density.

Lines 565-566: “Cell density was calculated based on the volume of culture required to capture 100,000 cells, at a flow rate of $35 \mu\text{L min}^{-1}$ (core size $16 \mu\text{m}$) using the flow cytometer.”

Figure 4C

- It would be easier for readers to understand if the name of the gene to be deleted was specifically written on line 112.

Thank you for the comment. We have now modified the sentences to read as follows:

Lines 115-116: “Consequently, we constructed single-, double-, and triple-gene deletion strains of *MFa1*, *MFa2*, and *STE3* genes (MTS001 to MTS007) and crossed them with the α -mating-type haploid BY4741 strain.”

Lines 120-123: “However, to avoid potential growth disparities stemming from pheromone-induced G_1 -arrest that could affect the stability of a microbial consortium, we selected the triple-gene deletion strain MTS007, which has the complete set of *MFa1*, *MFa2*, and *STE3* genes disrupted, as our sterile base strain.”

- In lines 142 to 144, the authors claim that modulating *HO* expression is typically unfeasible, but why not also state the reason?

Thank you for the comment. We explained in the succeeding sentence that the formation of a microbial consortium with mating-type heterogeneity is unfeasible through direct modulation of *HO* expression, as the replacement of the *HO* promoter will disrupt the unique spatiotemporal pattern of *HO* expression critical for maintaining the asymmetry in the mating types of the haploid mother and daughter cells. We have now modified the sentence for improved clarity.

Lines 146-150: “Notably, the formation of a consortium with mating-type heterogeneity is typically unfeasible through direct modulation of *HO* expression in the wild-type haploid *S. cerevisiae* strains. This is because replacing the *HO* promoter would disrupt the critical spatiotemporal pattern of *HO* expression necessary for establishing sexual asymmetry between the haploid mother and daughter cells.”

- In lines 215 to 216, the authors claim leaky *HIS3* expression, but the expression data is missing. The authors should be better to analyze transcript of *HIS3* gene.

Thank you for the suggestion. In this experiment, we aimed to study the effects of *SHE1-3* deletion on gene expression from the *HO* promoter. The *HIS3* gene was employed as the reporter gene, and a histidine drop-out plate was used to monitor its translation. The histidine drop-out plate was

highly selective: the control cell, BY4742, which lacks the *HIS3* gene, could not grow on this plate (Figure 2D, last row). Therefore, any cell growth on the selection plate unequivocally indicated *HIS3* translation, and by extension, its transcription. The new data has been added to Figure 2D, as shown below.

Additionally, we conducted a parallel experiment whereby the *HIS3* gene was substituted with the *GFP* gene and presented the data in Figure S3. As shown in Figure S3, green fluorescence in the *she3Δ* strain (MTS029) was significantly higher (p-value <0.003) compared to the two controls: BY4742 without the *GFP* gene, and the MTS030 strain with repressed *GFP* expression from the *GALI* promoter. Notably, the green fluorescence in the *she3Δ* strain (MTS029) was lower than that in the MTS028 strain with intact *SHE3* gene, indicating that the observed lowered fluorescence was due to leaky *GFP* expression from the *HO* promoter. Since both assays examined the translation product of the reporter genes, which necessitates prior transcription, we assert that these experiments adequately demonstrate leaky gene expression from the *HO* promoter.

We have added the following sentences to include the methods and the results of the fluorescence experiment.

Lines 218-222: “Instead, these incomplete inhibitions common to all deletion mutants likely resulted from leaky *HIS3* expression from the *HO* promoter. Notably, when the *HIS3* reporter gene was replaced with the *GFP* gene, similar leaky expression from the *HO* promoter was observed, resulting in green fluorescence approximately two-fold (p-value <0.003) higher than that of the BY4742 control and the strain MTS030 with repressed *GFP* expression from the *GALI* promoter (Figure S3).”

Line 517-522: “Strains MTS028-MTS030 (and the control BY4742) were similarly prepared and incubated in synthetic defined medium containing 2% glucose for the examination of leaky *GFP* expression from the *HO* promoter.”

Figure 2D

Figure S3

8. In lines 267-268, the authors mention that the varied responses to different inducer concentrations produced a diversity of population composition. However, the diversity would be dependent on the timing of induction by galactose and tetracycline. Therefore, the authors should explain how they decided on the induction timing.

Thank you for the comment. We selected 24 hours as the induction timing because, by the end of this duration, the cells would reach saturation (Figure S6). Our subsequent experiments showed no change in population composition for cells continuously incubated in the same inducing medium for more than 24 hours (up to three days; Figure 3G).

We have now added the following sentences in the Method section to explain this point.

Lines 531-532: “The induction timing was set to 24 hours because, by this duration, the cells would reach saturation (Figure S6).”

9. In lines 279-280, the authors speculate that this stability could result from the depletion of one or both inducers, but how about directly quantifying the concentration of the inducer?

Thank you for the comment. We postulated that the stability of the microbial consortium, when incubated in the same inducing medium for up to 3 days, could be due to depletion of either of the inducers as well as cell saturation. We demonstrated in subsequent experiments (Figure 3G) that changes in population composition occurred when the same induced cells were re-passaged and diluted (20-fold) in fresh inducing medium. Therefore, we made the following conclusion (Line 306-307) “*This indicates that reinduction is viable, provided inducers are replenished and the cell population is not saturated*”. Given that the inducer galactose is a carbon source consumed by *S. cerevisiae*, its concentration would drop as the cells grow, especially in the absence of glucose. This was demonstrated in a few studies earlier (<https://doi.org/10.1007/s00239-022-10079-9>; [https://www.cell.com/trends/genetics/abstract/S0168-9525\(21\)00256-0](https://www.cell.com/trends/genetics/abstract/S0168-9525(21)00256-0)). The absence of galactose rendered inactivation of the AND logic gate even in the presence of tetracycline (Figure 3B).

We have now modified the sentences as follows to improve clarity:

Lines 297-302: “Interestingly, this stability in population composition persisted even when the consortia were cultivated for an additional two days in the same inducing medium (Figure 3G). In addition to the depletion of one or both inducers, particularly galactose, which serves as a carbon source for the cells in the absence of glucose, we postulated that stability in this scenario could result from cell saturation after one day of cultivation, which could limit further cell divisions necessary for mating-type switching.”

10. Line 282 mentions cell division, but the authors should collect data on the number of cells. In the first place, it is dangerous to discuss cell density based solely on OD, as cell size may have changed due to genetic manipulation.

Thank you for your suggestion. We collected data on the number of cells and replotted Figure 4C (as shown in response to comment #4) as the percentage of α -mating type versus xylose production per cell.

11. When considering the application of the mating-type switching technology developed in this study, it is necessary to evaluate the cell proliferation ability of the constructed strain. Although this study collected data on the ratio of α -type and a-type, it would be better to also investigate the growth ability and doubling time of individual strains.

Thank you for the comment. Following your suggestion, we characterized the growth abilities of strain MTS019 (capable of switching mating type) and its α - and a-mating type derivatives, MTS031 α and MTS031a, respectively, which were unable to switch mating type. In addition, we characterized the growth abilities of the transformant derivatives of these strains harbouring either the pRSII416-XynII-XylA, pRSII416-MCp-XX or pRSII416-MFa2p-XX plasmids. The latter two plasmids were recently constructed: the former utilized P_{MC} to express both xylanolytic enzymes while the latter utilized the P_{MFa2}. The former was transformed into the MTS031 α strain and the latter into the MTS031a strain. The resulting transformant strains served as two single-strain monoculture controls for comparison with our consortia in xylose production. Our results indicated that strains MTS019, MTS031 α , MTS031a and their transformant derivatives all exhibited comparable growth rates, with mean doubling times ranging from approximately 81.5 to 89.8 minutes (p-value >0.5).

We have now provided this new data as Figure S6 and added the following sentences in the manuscript for discussion.

Lines 365-368: “Strains MTS019, MTS031 α , MTS031a, and their respective plasmid transformants exhibited comparable growth rates, with mean doubling times ranging from approximately 81.5 to 89.8 minutes (p-value >0.5). This indicated that there were no growth differences due to mating type or differential expression of *XYNII* and *XYLA* genes (Figure S6).”

The following sentences were added to the Method section to describe the methods used to characterize the growth abilities of the various strains.

Lines 498-506: “Overnight seed cultures of strains BY4742, MTS019, MTS031 α , MTS031a, MTS019 pRSII416-XynII-XylA, MTS031 α pRSII416, MTS031 α pRSII416-MCp-XX, and MTS031a pRSII416-MFa2p-XX were resuspended in synthetic defined medium or synthetic minimal medium lacking uracil containing 2% glucose to an initial OD₆₀₀ of 0.1. The samples were then loaded in triplicates of 100 μ L onto a 96-well microplate and subjected to a 20-hour kinetic analysis using a Synergy H1 Multi-Mode Reader (BioTek Instruments, Winooski, VT, USA). Cells were incubated at 30 °C, 807 cpm, with OD₆₀₀ measured at a 15-minute interval. Doubling time was calculated by dividing Ln(2) by the exponent of cell density versus time during the exponential phase (135 to 660 minutes). The mean doubling time was determined by averaging the doubling times of the three independent replicates.”

Figure S6

12. On line 353, the authors claim that determining the optimal enzyme ratios, but they should explain this application example more specifically. Will the technique of this study be an effective approach when optimizing the ratio of what enzymes?

Thank you for the comment. We previously discussed this point in the application section in lines 349-355:

“Notably, higher xylose production per cell was observed in consortia with a balanced proportion of haploids of opposite mating types (40 to 60% α-mating type). Given the matched promoter strengths of P_{MC} and P_{Mfa2}, this indicates that the XynII and XylA enzymes are equally efficient, necessitating a balanced presence for optimal productivity. Should one enzyme exhibit lower activity than the other, an increased quantity of the less active enzyme and its corresponding mating type would be necessary. This offers versatile tools for tuning and determining the ratio of pathway enzymes to achieve optimum flux.”

For clarity, we have added the following sentences to explain this application more specifically in the conclusion section.

Lines 410-413: “The ability to generate microbial consortia with diverse population compositions holds significant promise in synthetic biology, particularly for optimizing bioproduction. By distributing biosynthetic enzymes into two parts expressed under haploids of opposite mating types, the optimal enzyme ratio can be inferred by observing the population composition needed for maximum productivity.”

Reviewer #2 (Remarks to the Author):

In this manuscript, entitled “Tunable cell differentiation via reprogrammed mating-type switching”, the authors present a well-written work with clever rational design features and a clear logic to it. By installing logic gates, they achieve a high degree of control over inducible mating-type switching in *S. cerevisiae*, which they then apply to division of roles in relation to biotechnology. Indeed, the topic is highly relevant and contributes to an active field of research. However, several minor and major issues need to be resolved prior to publication:

Minor comments:

1. Resolution appears to be low for flow cytometry plots. The sizes of flow cytometry plots are too small in 1D and 2A&E, and 3A. New subpopulations (plural) relative to Fig. 1D occur in the MAT α gate in 2A, 2E, and 3A, and, presumably due to poor reporter-performance of the MF α 1 promoter driving GFP, fluorescence profiles overlap – which is particularly obvious in the subpanel of Fig. 1D, where the subpopulation of this reporter is in Q4, and not in Q3 as otherwise would have been expected. Hence, I currently disagree with the line 656 statement “Expression of GFP and mCHERRY was mutually exclusive”.

Thank you for the comments. We have now modified Fig 1D, 2A, 2E, and 3A to improve the resolution of the flow cytometry plots. In addition, we have provided the enlarged version of the flow cytometry plots as supplementary information (Figures S1 & S2).

The α -mating type shown in Figure 1D refers to strain MTS008 α , a derivative of the sterile strain MTS007, which was unable to switch mating type due to a recessive *ho* gene. Consequently, the population derived from this strain comprised homogeneously α -mating type haploids. The subpopulation observed in Q4 of the flow cytometry plot of this strain (Figures 1D and S1A) represents α -mating type haploids with no green fluorescence, notably at a level comparable to the BY4742 control lacking any fluorescent reporter constructs (the grey population in Figure S1C). This subpopulation is not a result of the activity of the $P_{MF\alpha 2}$ -*mCHERRY*- $T_{MF\alpha 2}$ reporter construct, as it was similarly observed in strain MTS027 α (sterile, did not switch mating type, α -mating type haploids) lacking the mCherry reporter construct (Figure S1D). Additionally, it was unrelated to the performance of $P_{MF\alpha 1}$ -*GFP*- $T_{MF\alpha 1}$ reporter construct, as the same subpopulation was observed when *GFP* was expressed under the strong *GALI* promoter in the presence of 2% galactose (strain MTS030; Figure S1F). Thus, we conclude that this subpopulation is a part of the α -mating type haploid population exhibiting no green fluorescence. To improve clarity, we have now removed the BY4742 control from the “Mixture” flow cytometry plot in Figure 1D. Instead, the original plot including the BY4742 control is provided as supplementary information under Figure S1C.

Figures 2A, 2E, and 3A (and Figure S2) are flow cytometry plots of strain MTS009, MTS018, and MTS019, respectively, which possessed a dominant *HO* gene and were capable of switching mating type. The other subpopulation observed in Q3 of their fluorescence scatter plots, which exhibited high green and red fluorescence, are deduced to be haploids that recently switch from a-to- α mating type. This inference stems from the differential kinetics of mCherry and GFP. Despite switching from a-to- α mating type, the mCherry remains in the cells due to its long half-life. Coupled with the shorter maturation time of GFP, these result in the new subpopulation exhibiting high green and red fluorescence due to the concurrent presence of both fluorescent proteins in the cells. This scenario is unlikely when the haploids switch from α -to-a mating type, as the degradation of GFP would be faster than the maturation of mCherry. The presence of this new subpopulation especially in strains with leaky *HO* expression, such as strain MTS009 (in the absence of galactose inducer; Figures 2E and S2B), supports the idea that it represents recently switched cells.

The statement “*Expression of GFP and mCHERRY was mutually exclusive*” pertains to the observation that *GFP* expression under the *MF α 1* promoter was specific to the α -mating type, while *mCHERRY* expression under the *MF α 1* promoter was exclusive to the a-mating type. This was supported by the absence of green fluorescence in the a-mating type strain MTS008 α , which was unable to switch mating type (Figure 1D and S1C). The green fluorescence observed in this strain was comparable to that observed in the BY472 control (due to autofluorescence) and the subpopulation of the α -mating type strain MTS008 α (unable to switch mating type) in Q4. Conversely, a red fluorescence higher than the BY4742 control, but lower than the a-mating type strain MTS008 α , was observed in the MTS008 α . However, this fluorescence was unrelated to *mCHERRY* expression but was instead due to spillover signals from GFP. Notably, the flow cytometer Accuri C6 utilized a blue laser to excite GFP and mCherry at a wavelength of 488 nm.

The emission spectrum of GFP fell within the filtering ranges of both the FL1 (520/30 nm) and FL4 (610/20 nm) channels, and so its signal could be readily detected by both channels, albeit sub-optimally by the latter. In our experiments, the FL1 channel was used to detect and quantify GFP expression, while FL4 was used for the mCherry. This suggests that the red fluorescence detected in the MTS008 α strain originated from GFP. Further support was drawn from the flow cytometry plot of another α -mating type strain, MTS027 α , which did not switch mating type. This strain lacked the $P_{MF\alpha 2}$ -*mCHERRY*- $T_{MF\alpha 2}$ reporter construct, and thus did not express mCherry. A higher-than-control red fluorescence was similarly observed in this strain. Likewise, a similar observation was reported for strain MTS030 expressing *GFP* from a *GAL1* promoter and lacked a *mCHERRY* reporter construct. These findings confirmed that the above-control red fluorescence observed in the α -mating type haploids was due to spillover signal from GFP. Therefore, we concluded that the expression of both *GFP* and *mCHERRY* was mutually exclusive.

We have now modified or added the following sentences to address the points above.

Lines 131-140: “As illustrated in Figure 1D and S1, GFP was exclusively expressed in the α -mating type haploids (MTS008 α), and mCherry in the a-mating type haploids (MTS008a). Notably, red fluorescence higher than the BY4742 control but lower than the MTS008a strain was observed in the MTS008 α strain, which was later found to be caused by spillover signal from GFP detected by the red fluorescence channel (FL4). Both MTS008 α and MTS008a strains formed distinct populations with minimal overlap on a fluorescence scatter plot. However, direct quantification of population composition by evaluating the proportion of cells in Q3 and Q4 was challenging due to the presence of a small subpopulation of α -mating type haploids (approximately 2.63% based on 144 samples) that overlapped with the a-mating type haploids in Q4.”

Lines 232-239: “Notably, a subpopulation was observed in Q3 of the fluorescence scatter plot, representing haploids exhibiting high green and red fluorescence. This subpopulation was also present in strain MTS009 after galactose induction, although less prominently. Considering the differential kinetics of the mCherry and GFP fluorescent proteins, we inferred that this subpopulation comprised haploids that recently switched from a-to- α mating type. Due to a longer half-life, mCherry remains in the α -mating type haploids, leading to its concurrent presence with GFP, which has a shorter maturation time. This scenario is unlikely when the haploids switch from α -to-a mating type, as the degradation of GFP is faster than the maturation of mCherry.”

Figure S1

Figure S2

- It is unclear to me how the authors considered the MAT α sub-populations in the handling of their flow cytometry analyses throughout – particularly the ones that obviously float into the MAT α gate (Fig. 1D “mixture”). Please clarify in detail how you analyzed your flow cytometry data and explain how you took the overlapping subpopulations into account.

Thank you for the comment. As previously discussed in our responses above, we identified a small subpopulation of α -mating type haploids exhibiting no green fluorescence. Notably, this subpopulation overlapped with the a-mating type haploids in Q4 of a flow cytometry plot. To address this, we quantified and averaged the percentage of cells falling within Q3 and Q4 for the α -mating type strain, MTS008 α , which was unable to switch mating type. Our findings revealed that, on average, 97.37% and 2.63% of α -mating type haploids fell within Q3 and Q4, respectively (n=144). Based on this data, we assumed that 97.37% of cells in Q3 represent 100% α -mating type haploids. For all subsequent experiments, we determined the percentage of α -mating type haploids by dividing the percentage of cells in Q3 by 97.37%. The percentage of a-mating type haploids was calculated by subtracting the calculated percentage of α -mating type haploids from 100%.

We now added the following sentences to explain this in the manuscript.

Lines 136-142: “Both MTS008 α and MTS008a strains formed distinct populations with minimal overlap on a fluorescence scatter plot. However, direct quantification of population composition by evaluating the proportion of cells in Q3 and Q4 was challenging due to the presence of a small subpopulation of α -mating type haploids (approximately 2.63% based on

144 samples) that overlapped with the a-mating type haploids in Q4. To account for this, we estimated the proportion of α -mating type haploids in a population by dividing the percentage of cells in Q3 with 97.37% and determined the proportion of a-mating type haploids by subtracting the calculated percentage of α -mating type haploids from 100%.

3. Line 131 and 133: Please change “GFP” and “mCHERRY” to italics as you are referring to genes here.

Thank you for the comment. We have italicized “*GFP*” and “*mCHERRY*” as suggested.

4. Line 49: Consider changing “(...) different cells (...)” to “(...) different cell types (...)”.

Thank you for the comment. We have changed “different cells” to “different cell types” as suggested.

5. Fig. 2E and 3A do not properly illustrate the point that GFP or mCherry is expressed in MAT α or MATa cells, respectively. Please consider changing the repression mark on mCherry to an arrowhead and mark the arrows going to GFP or mCherry by e.g., “ α -only” or “a-only”, respectively.

Thank you for your suggestions. We have revised Figure 2A, 2E, and 3A, as shown below.

Figure 1A

Figure 1E

Figure 2A

6. Fig. 3G: “100” percent should be formatted to be in one line.

Thank you for the comment. We have formatted “100’ as suggested.

7. Please mention the severe limitation of strong promoter availability specific to the two mating types. It can be done in line 358 in connection with the claim “versatile toolkit”.

Thank you for the comment. We have now added the following sentences to address the limitations of strong mating-type-specific promoters.

Lines 414-419: “Additionally, apart from modulating the population composition, the ratios of these enzymes could be adjusted using the seven mating-type-specific promoters characterized in this study. While their varying expression strengths offer advantages in fine-tuning enzyme expression, the moderate expression strengths of these mating-type specific promoters may pose limitations in certain contexts. Further investigation is therefore warranted to explore engineering strategies that can enhance the expression strengths of these promoters while preserving their mating-type specificity.”

8. Please further discuss both advantages and disadvantages by your isogenic strain approach in relation to other recent demonstrations on division of labor in synthetic yeast co-cultures - e.g.,

Peng et al. (<https://www.nature.com/articles/s41564-023-01596-4>). This can be done in connection with lines 346-348 where, however, human intervention is not impractical.

Thank you for the suggestion. We have now added the following discussion on the advantages and disadvantages of our approach in relation to the other study.

Lines 392-408: “We posit that the approach delineated in this study represents a significant advancement in the development of synthetic microbial consortia. Our approach offers the advantage of utilizing isogenic consortium members derived from the same strain, thereby enhancing stability as the members can be readily differentiated from one another. Indeed, instability arising from differential cell growth, incompatibility and competition has been a major issue hampering synthetic microbial consortia. A recent study addressed this challenge by fostering interdependency among consortium members. Specifically, the authors engineered and deployed cross-feeding auxotrophic and overexpression yeast strains to construct two- and three-member consortia and demonstrated their stability and effectiveness in enhancing resveratrol production through division of labor. While our present study focused on constructing a two-member consortium, our approach holds potential for further adaptation. We envisage the incorporation of additional homothallic yeast strains and their combinations to expand the genetic diversity, complexity, and functionality of the microbial consortia formed. Moreover, the inducibility of our approach eliminates the need for manual mixing various consortium members, enabling microbial consortia formation and cooperativity in environments where direct human intervention is impractical. Future iterations could explore using alternative inducible promoters to regulate *SHE3* expression to facilitate the formation of microbial consortia and cooperativity under specific environmental conditions.”

9. I did not find the raw data e.g., for flow cytometry and LC-MS in the original submission.

We have now provided the raw data as source data, and representative HPLC chromatograms in Figure S5.

Major comments:

10. References appear to be out of chronological order throughout the manuscript: Line 41: Reference 7 appears to be the first cited work. Line 79 to line 106: REFs 23+24 jump straight to REFs 41+42. Line 303 introduces REFs 25-27 (following REF 55 in line 244). Etc. Please make sure to correct it.

Thank you for the comment. We have amended the references according to chronological order.

11. Line 162: Galactose-induction is binary, unless *GAL2* is deleted (<https://doi.org/10.1074/jbc.M512317200>). The “titratable” experiments in Fig. 3D and 3G should be done again in a *gal2Δ* genetic background, particularly in the view of claiming “Further alterations in galactose concentration from 0.5 to 8.0% did not significantly change the population composition, which we conjectured reached equilibrium with the α -to- α mating type ratio maintained at approximately 0.14”. This experiment potentially makes the system more tunable.

Thank you for the suggestion. The claim that “*Further alterations in galactose concentration from 0.5 to 8.0% did not significantly change the population composition, which we conjectured reached equilibrium with the α -to- α mating type ratio maintained at approximately 0.14*” pertains to the performance of the YES logic gate based on direct modulation of *HO* expression using a

GALI promoter (strain MTS009; Figure 2B). This assertion was made based on the observation that despite adjustments in galactose concentration, mating type compositions remained relatively comparable (slope=0.0057; $R^2=0.84$), indicating saturation of *HO* expression from the strong *GALI* promoter.

In contrast, Figures 3D and 3G depict the performance of the AND logic gate based on indirect modulation of *HO* expression (strain MTS019). In this case, we observed a linear decrease in the proportion of α -mating type when adjusting the galactose concentration from 0.5 to 8.0%, in the presence of either 12.5 $\mu\text{g mL}^{-1}$ (slope= -0.0397; $R^2=0.94$) or 25.0 $\mu\text{g mL}^{-1}$ (slope= -0.0130; $R^2=0.83$) tetracycline. This suggests tunability of the AND logic gate through variations in galactose and tetracycline concentrations, as evidenced by the diverse population compositions produced (Figure 3E).

We have now provided the slope and R^2 values to ease comparison of the linearity between the constructs above.

Lines 165-168: “Further alterations in galactose concentration from 0.5 to 8.0% did not significantly change the population composition (slope=0.0057; $R^2=0.84$), which we conjectured reached equilibrium with the α -to-a mating type ratio maintained at approximately 0.14.”

Lines 282-287: “Adjusting the galactose concentration from 0.5 to 8.0% yielded similar linear dose responses, with a more pronounced change occurring in the medium with 12.5 $\mu\text{g mL}^{-1}$ (slope= -0.0397; $R^2=0.94$) than with 25.0 $\mu\text{g mL}^{-1}$ (slope= -0.0130; $R^2=0.83$) tetracycline (Figure 3D). These varied responses to different inducer concentrations demonstrated the tunability of the AND logic gate, which produced a diversity of population compositions with α -to-a mating type ratios ranging from 0.5 to 50.8 (Figure 3E).”

12. Fig. 4B: Please show expression levels by GFP in both mating types for all promoters, and in particularly for pMC, as it is a new hybrid promoter. This all talks into system control, and consequently the division of “role”.

Thank you for your suggestion. We further characterized *GFP* expression by $P_{MF\alpha 1}$ and P_{MC} in haploids of opposite mating types, and the findings were presented in Figure S4. The results indicated that P_{MC} exhibited approximately half the expression strength of $P_{MF\alpha 1}$ and was predominantly active in the α -mating types.

We have added the following sentences in the manuscript to describe this finding.

Lines 330-337: “However, none of the native promoters formed a perfectly matched pair, prompting the creation of a hybrid α -mating-type-specific promoters, P_{MC} (Figure S4). This hybrid promoter was created by replacing the TATA box-containing sequence downstream of the *MFa1* promoter (-275 to -1 bp) with that from the *CYCI* promoter (-252 to -1 bp), while keeping the upstream sequence (-1000 to -276 bp) containing the P’Q elements essential for α -mating-type-specific expression intact. With an expression strength half that of the *MFa1* promoter, this newly created hybrid promoter formed a matching pair with the α -mating-type-specific promoter, $P_{MF\alpha 2}$.”

Figure S4

13. Fig. 4C: It is necessary to show a “mono-culture” control for the conversion of xylan to xylose. Please benchmark to co-display on a strain that does not undergo mating-type switching (cited as REFs 25-27), by using strong constitutive promoters (TDH3 or CCW12). Importantly, potential suboptimal performance in your system as compared to a mono-culture does not in my opinion prevent publication, but rather serves to help potential users of your technology to understand its advantages and limitations, and hence to plan their experimental work accordingly. Whatever the outcome, please include it in your discussion in the end.

Thank you for your suggestion. We have included “mono-culture” controls as advised and presented the data in Figure S7. Specifically, we constructed both α - and a-mating type derivatives of strain MTS019 that were unable to switch. This was accomplished by replacing the dominant *HO* gene in strain MTS019 with the recessive *ho* gene. The resulting strains, MTS031 α (α -mating type) and MTS031a (a-mating type), were transformed with plasmid derivatives of pRSII416-XynII-XylA. These plasmids, denoted as pRSII416-MCp-XX and pRSII416-MFa2p-XX, utilized either the P_{MC} or P_{MFa2} to regulate the expression of both the *XYNII* and *XYLA* genes. This design rendered *XYNII* and *XYLA* expression in the α -mating type haploid for the former and in the a-mating type haploid for the latter.

The “mono-culture” controls were represented by the resulting transformant strains, MTS031 α pRSII416-MCp-XX (α -mating type) and MTS031a pRSII416-MFa2p-XX (a-mating type). We compared their xylose production to those of consortia established by strain MTS019 pRSII416-XynII-XylA. As shown in Figure S7, xylose production per cell by the “mono-culture” controls were approximately four folds higher than the maximum observed in the consortia. We elaborated on this by adding the following discussion:

Lines 357-368: “To compare the performance of microbial consortia with the typical single-strain monoculture system in xylose bioproduction, we constructed both the α - and a-mating type derivatives of strain MTS019, which were unable to switch mating type. This was accomplished by replacing the dominant *HO* gene in strain MTS019 with the recessive *ho* gene. The resulting strains, MTS031 α (α -mating type) and MTS031a (a-mating type), were transformed with plasmid derivatives of pRSII416-XynII-XylA. These plasmids, denoted as pRSII416-MCp-XX and pRSII416-MFa2p-XX, utilized either the MC or MFa2 promoter to

regulate the expression of both the *XYNII* and *XYLA* genes (Figures S7A and S7B). This design rendered *XYNII* and *XYLA* expression in the α -mating type MTS031 α strain for the former, and in the a-mating type MTS031a strain for the latter. Strains MTS019, MTS031 α , MTS031a, and their respective plasmid transformants exhibited comparable growth rates, with mean doubling times ranging from approximately 81.5 to 89.8 minutes (p-value >0.5). This indicated that there were no growth differences due to mating type or differential expression of *XYNII* and *XYLA* genes (Figure S6).”

Lines 370-379: “As shown in Figure S7C, xylose production per cell by the MTS031 α pRSII416-MCp-XX and MTS031a pRSII416-MFa2p-XX strains were approximately 4-fold higher (p-value <0.05) than the maximum observed in the microbial consortia established by strain MTS019 pRSII416-XynII-XylA. Although the cell surface display of XynII and XylA enzymes bypass the need of membrane crossing for intermediate exchange, the proximity of these enzymes on the same cell likely explains the higher productivity of the monocultures. Nonetheless, we believe that our system will be advantageous in scenarios where the expression of a large biosynthetic pathway in a single chassis is limiting, making cooperativity within a microbial consortium beneficial. The differential expression of such biosynthetic pathway would be greatly facilitated by leveraging the collection of mating-type-specific promoters characterized in this study.”

The following sentences were added to the Methods section to describe the methods used in these experiments.

Line 476-478: “Plasmids pRSII416-MCp-XX and pRSII416-MFa2p-XX were constructed by replacing the P_{MFa2} -*XYLA*- T_{CYC1} and P_{MC} -*XYNII*- T_{ADHI} of plasmid pRSII416-XynII-XylA, with P_{MC} -*XYLA*- T_{CYC1} and P_{MFa2} -*XYNII*- T_{ADHI} , respectively (Figures S7A and S7B).”

Line 557-562: “For xylose bioproduction using single-strain monoculture system, overnight cultures of transformant strains MTS031 α pRSII416-MCp-XX and MTS031a pRSII416-MFa2p-XX, were similarly inoculated to an initial OD600 of 0.2 in synthetic minimal medium lacking uracil containing 2% glucose and incubated with 1% beechwood xylan (Megazyme) at 30 °C, 225 rpm for 24 hours. Strains MTS031 α pRSII416, MTS031 α pRSII416-MFa2p-XX, MTS031a pRSII416 and MTS031a pRSII416-MCp-XX were included as controls.”

Figure S7

Reviewers' Comments:

Reviewer #1:

Remarks to the Author:

The authors answered the majority of the reviewers' criticisms satisfactory.

Reviewer #2:

Remarks to the Author:

Thank you for your improvements according to my previous comments. What remains is relatively minor:

Point 1. Regarding spillover: I suggest performing compensation on your flow cytometry data, which is clearly needed based on the shape in Q3 (e.g., Fig. 1D & S1F).

Point 13. OK, please make sure that Figure S6 (line 368) is referred to before introducing Figure S7 (line 363).

Responses to reviewers' comments for manuscript submission NCOMMS-24-12398A

We thank the reviewers for their comments. The point-by-point response to the comments is provided below.

Reviewer #1 (Remarks to the Author):

The authors answered the majority of the reviewers' criticisms satisfactory.

We thank you once again for your time and constructive suggestions, which we believe have greatly improved the quality of our manuscript.

Reviewer #2 (Remarks to the Author):

Thank you for your improvements according to my previous comments. What remains is relatively minor:

We are grateful for your constructive suggestions, which have greatly helped us improve our manuscript.

1. Point 1. Regarding spillover: I suggest performing compensation on your flow cytometry data, which is clearly needed based on the shape in Q3 (e.g., Fig. 1D & S1F).

Thank you for your suggestion. We believe that compensation is not needed in our case, as our primary goal is to separate haploids of opposite mating types based on differential expression of GFP and mCherry fluorescent proteins. Although spillover from GFP contributes to background red fluorescence in α -mating type haploids, it does not affect their separation from a-mating type haploids. The α -mating type haploids are distinguished as cells mainly in Q3, while a-mating type haploids are cells in Q4. We also recognize that compensation can sometimes lead to overcompensation issue, which might impact data analysis ([https://onlinelibrary.wiley.com/doi/10.1002/1097-0320\(20011101\)45:3%3C194::AID-CYTO1163%3E3.0.CO;2-C](https://onlinelibrary.wiley.com/doi/10.1002/1097-0320(20011101)45:3%3C194::AID-CYTO1163%3E3.0.CO;2-C); <https://doi.org/10.1016/B978-0-12-813776-5.00027-3>). Therefore, we have decided to use the raw flow cytometry data for our analysis.

2. Point 13. OK, please make sure that Figure S6 (line 368) is referred to before introducing Figure S7 (line 363).

Thank you for your suggestion. We have swapped the sequences of Figures S6 and S7. The former Figure S7 is now Figure S6 and is introduced first in the text, followed by the former Figure S6, which is now Figure S7.